# Multiomics reveals persistence of obesity-associated immune cell phenotypes in adipose tissue during weight loss and weight regain in mice

Matthew A. Cottam[1,3], Heather L. Caslin[1,3], Nathan C. Winn [1] & Alyssa H. Hasty [1,2 ✉]

Within adipose tissue (AT), immune cells and parenchymal cells closely interact creating a complex microenvironment. In obesity, immune cell derived inflammation contributes to insulin resistance and glucose intolerance. Diet-induced weight loss improves glucose tolerance; however, weight regain further exacerbates the impairment in glucose homeostasis observed with obesity. To interrogate the immunometabolic adaptations that occur in AT during murine weight loss and weight regain, we utilized cellular indexing of transcriptomes and epitopes by sequencing (CITEseq) in male mice. Obesity-induced imprinting of AT immune cells persisted through weight-loss and progressively worsened with weight regain, ultimately leading to impaired recovery of type 2 regulatory cells, activation of antigen presenting cells, T cell exhaustion, and enhanced lipid handling in macrophages in weight cycled mice. This work provides critical groundwork for understanding the immunological causes of weight cycling-accelerated metabolic disease. For further discovery, we provide an open-access web portal of diet-induced AT immune cell imprinting: https://hastylab.shinyapps.io/MAIseq.

[1] Department of Molecular Physiology and Biophysics, Vanderbilt University School of Medicine, Nashville, TN, USA. [2] VA Tennessee Valley Healthcare System, Nashville, TN, USA. [3]These authors contributed equally: Matthew A. Cottam, Heather L. Caslin. ✉email: alyssa.hasty@vanderbilt.edu

Obesity affects more than 650 million adults worldwide and is associated with nearly every leading cause of death, including cardiovascular disease, diabetes, and several types of cancer[1,2]. In lean adipose tissue (AT), regulatory type 2 immune cells contribute to tissue homeostasis[3–6]. In contrast, weight gain promotes the infiltration of circulating immune cells, and tissue resident cells polarize towards a type 1 pro-inflammatory phenotype. These inflammatory changes contribute to metabolic dysfunction – promoting lipolysis, fibrosis, and insulin resistance.

Weight loss (WL) is known to improve metabolic outcomes associated with obesity. However, low success rates and failure to maintain lost weight are common, with recent studies reporting that most individuals (>60%) regain weight within a few years[7–10]. Importantly, weight cycling (WC) – the repeated process of gaining and losing weight – further increases risk for developing diabetes and cardiometabolic diseases in humans[11–13]. We generated a mouse model of WC using an alternating high fat diet (HFD) and low fat diet (LFD) feeding paradigm[14]. These mice display worsened glucose tolerance compared with obese mice, despite similar body weight, fat mass, and total time on the HFD. As assessed by flow cytometry, metabolic dysfunction was associated with increases in AT T cell populations, but not macrophages, consistent with other models of WC[15].

With obesity, cells can upregulate type 1 and type 2 transcriptional profiles and metabolic pathways simultaneously[16,17]. Moreover, recent single-cell RNA-sequencing (scSeq) studies have revealed multiple distinct AT immune cell populations and subsets, such as lipid-associated macrophages (LAMs), which are unique from traditional macrophage polarization states[17–19]. scSeq allows for the interpretation of changes to immune populations and unbiased gene expression across the entire immune landscape simultaneously. Further developments, such as Cellular Indexing of Transcriptomes and Epitopes by sequencing (CITE-seq) and cell hashing, offer opportunities to interrogate surface protein repertoire and include biological replicates during scSeq experiments[20,21].

To date, no studies have thoroughly characterized AT immune populations following WC by a comprehensive technique like scSeq, and many published scSeq datasets are difficult to explore further without bioinformatics expertise. Moreover, most reports are limited to single cell types and fail to capture the dynamic changes across multiple immune cell types. Due to the associated costs, personnel time, and necessary bioinformatics training, there remains a critical need to improve accessibility of high-resolution data provided within scSeq datasets. Therefore, the aims of this study were two-fold: (1) To comprehensively map the changes in adipose immune populations with obesity, WL, and WC and identify key links between WC and metabolic disease and (2) To provide a freely accessible resource for hypothesis generation and testing for the scientific community in a novel dataset that spans four distinct physiological states of AT. We report that obesity-associated inflammatory changes such as T cell exhaustion, antigen presentation, lipid handing, and inflammation persist following WL and worsened with weight regain in male mice, suggesting a memory-like immunological imprinting that may contribute to WC-accelerated metabolic disease.

## Results
**Diet-induced WC exacerbates glucose intolerance in male mice.** We used 9-week bouts of HFD and LFD feeding to generate models of obesity, WL, and WC in male mice (Fig. 1a)[14]. This model is robust, as shown across 6 cohorts (totaling 252 mice). Changes in weekly body mass and energy intake were tightly linked to prescribed diets (Fig. 1b, c). Cumulative energy intake

was not different between obese, WL, and WC groups (Fig. 1d). After 26-weeks on diet, obese and WC mice had identical lean and fat mass, which was elevated compared to lean and WL mice (Fig. 1e). WL animals had greater lean mass but no differences in fat mass compared to lean controls. Obese mice had impaired glucose tolerance by intraperitoneal glucose tolerance tests (ipGTT) compared to lean mice, which was further worsened by WC (Fig. 1f). Fasting insulin concentrations were not different between obese and WC animals (Obese, $5.2 \pm 0.8$ ng/ml and WC, $5.1 \pm 0.9$ ng/ml, $p = 0.9$, $n = 8$/group). A small improvement in glucose tolerance was observed in WL mice compared to lean control mice. Fat free tissue comprises the bulk of insulin-stimulated glucose disposal and is positively associated with postprandial glucose clearance[22,23]. Given that lean mass was greater in WL versus lean mice and WL animals had greater glucose clearance than the lean group, lean tissue mass was covaried against GTT AUC. The decrease in glucose clearance in WL compared to lean animals manifested after statistically accounting for differences in lean mass (estimated marginal means: Lean AUC, $19,865 \pm 915$; WL AUC, $16,750 \pm 1125$, $p < 0.05$). Effects of HFD feeding and weight loss (between 3, 9, and 18 weeks of the intervention) on body composition and glucose tolerance are reported in the data supplement (Supplementary Fig. 1).

Subcutaneous AT (sAT) and liver weights were similar between lean vs. WL and obese vs. WC mice. However, WL mice had reduced epididymal (eAT) mass compared to lean mice, and WC mice had increased eAT mass compared to obese mice (Fig. 1g). While sAT and liver mass as a proportion of total body mass were similar between lean vs WL mice and obese vs WC mice, the percentage of eAT mass was significantly higher in WC vs obese mice, albeit very modest (Fig. 1h). Immuno-fluorescence staining for perilipin-1, a lipid droplet membrane protein, revealed a slight reduction in lipid droplet diameter in eAT of WL compared to lean mice, while no difference was observed between obese and WC mice (Fig. 1i, j). The corresponding body mass, food intake, tissue mass, and ipGTT for the subset of 4 male mice per group included in subsequent scSeq experiments is shown in Supplementary Fig. 2.

We also determined whether WC worsened metabolic control in female mice. At the end of the 27-week study, body weight, lean mass, and fat mass were greater in both obese and WC females than lean controls, whereas no differences were detected between obese vs WC groups. In contrast to male mice, weight cycling did not significantly exacerbate glucose intolerance in female animals (i.e., glucose AUC); however, there was a modest delay in glucose clearance during the glucose excursion between WC and obese females (Supplementary Fig. 3).

Together, these data demonstrate that our mouse model provides a robust representation of WC-accelerated metabolic disease in male mice, which were used for all subsequent experiments.

**Multimodal single-cell sequencing highlights the diversity of adipose immune cells.** To profile the immune landscape across lean, obese, WL, and WC groups, we performed droplet-based scSeq with CITE-seq, whereby oligo-conjugated antibodies specific for surface protein targets are simultaneously sequenced with endogenous mRNA. CITE-seq antibodies were used to confirm and improve cell annotation as follows: T cells (CD3, CD4, CD8α, TCR γ/δ), NK cells (NK1.1), B cells (CD19), myeloid cells (CD11b), macrophages (FCγR1, MAC2/GAL3), DCs (CD11c), and neutrophils (CD39) as well as co-stimulation and activation/inhibition markers (CD279/PD-1, TIGIT, CD44, CD80, CCR7/CD197) (Supplementary Table 1). We also used cell hashing via

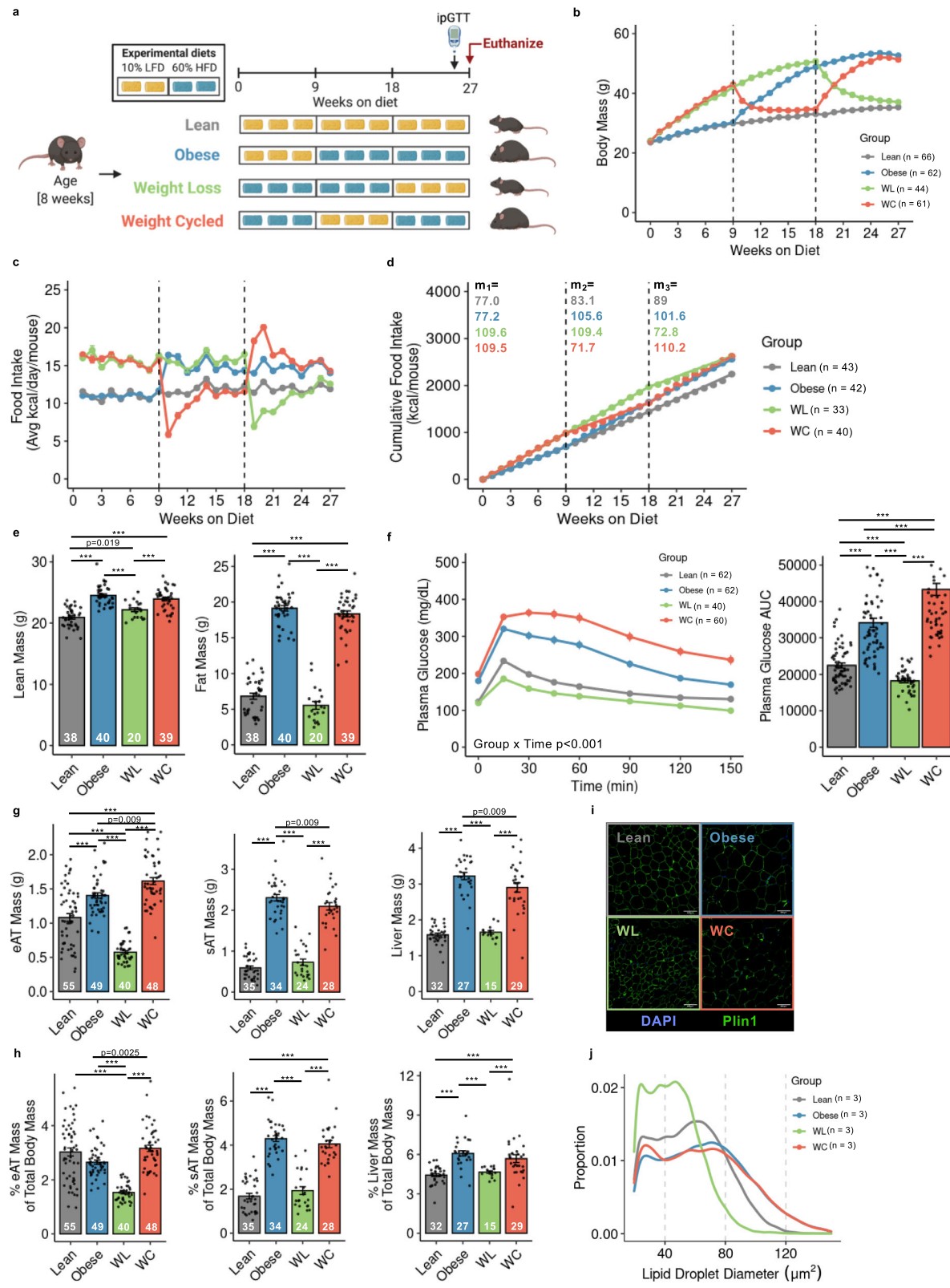

antibodies conjugated with unique oligos targeting ubiquitous surface proteins to pool and index 4 biological replicates per group. Cells were isolated via collagenase digestion, CD45+ magnetic enrichment, and FACS for viability before preparation for sequencing using the 10X Chromium platform (Fig. 2a). CITE-seq antibodies were validated by comparing sequenced protein expression levels to gene expression levels for matched targets (i.e. CD4 vs. *Cd4*), largely exclusive targets (i.e. CD4/ *Cd4* vs. CD19/*Cd19*), and for commonly associated targets (i.e. CD4/ *Cd4* vs. CD3ε/*Cd3e*) (Supplementary Fig. 4). Importantly, biological replicates are frequently ignored in scSeq experiments, and we sought to address this concern using cell hashing antibodies. Cell types were well represented in all biological replicates without any major outliers driving our interpretation and cell

**Fig. 1 Mouse models of lean, obese, weight loss (WL), and weight cycling (WC). a** Schematic of dietary approaches to generate WL and WC mice using 10% low fat diet (LFD) and 60% high fat diet (HFD). **b** Body mass over time measured weekly with diet switch indicated by dashed lines. **c** Food intake over time measured weekly. **d** Cumulative food intake measured throughout the duration of the studies with slope (m) for each 9-week segment indicated. **e** Lean and fat mass measured by nuclear magnetic resonance. **f** Blood glucose during an intraperitoneal glucose tolerance test (ipGTT) dosed at 1.5 g dextrose/kg lean mass one week prior to the end of the study and area under the curve (AUC) for ipGTT. **g** Tissue mass for epididymal adipose tissue (eAT), subcutaneous adipose tissue (sAT), and liver and **h** tissue mass as percentage of body mass at sacrifice. **i** Representative imaging of Perilipin-1 (Plin1) and 4′,6-diamidino-2-phenylindole (DAPI) immunofluorescence for lipid droplet size. (**j**) Distribution of lipid droplet size. For diet groups, gray = lean, blue = obese, green = WL, orange = WC. Pairwise two-tailed Student's *t*-tests with Bonferroni correction for multiple comparisons were used to compare groups for body composition, tissue mass, and ipGTT AUC and two-way ANOVA was used to compare groups for ipGTT; significant *p* values shown or ***$p_{adj}$ < 0.001. Data is plotted as mean ± SEM. Sample size (mouse per group; *n*) is indicated in corresponding figure legends or by white text at the bottom of each histogram. Figure 1a was created with Biorender.com.

classification (Supplementary Fig. 5a), and all data shown are normalized to the number of cells analyzed per mouse to correct for differences in total cell numbers.

Across the four sample groups (16 mice), a total of 33,322 cells that met strict quality control metrics (see Methods section) were retained and integrated (Supplementary Fig. 5b). Large cell clusters generated using low resolution nearest-neighbor clustering were first annotated using differentially expressed protein and gene signatures. Further subcluster annotation was conducted by subsetting broad cell type clusters. A heatmap of the top 5 genes associated with each cell type subcluster is shown in Supplementary Fig. 6. The resulting dimensional reduction with cells colored by subcluster is shown in Fig. 2b. A subset of genes associated with specific subclusters are highlighted in Fig. 2c. Antibody-based protein measurements for highly expressed surface proteins were also robust for cell type identification (Fig. 2d). Finally, a dendrogram was produced using the 2000 most highly variable genes to elucidate broad relationships between cell subclusters (Fig. 2e). Cell cluster designations were confirmed by published gene markers and are used throughout the rest of this manuscript to interrogate group differences by diet.

**Obesity-associated immune cell phenotypes are confirmed by single cell sequencing.** To validate our scSeq dataset, we compared cells isolated from obese male mice to those from lean male mice, as these differences are well documented. To explore how relative frequency of immune cells changed during obesity, we grouped cells into metacells (a group of cells connected by proximity to a representative index cell in our dimensional reduction) and compared abundance by diet group (Fig. 3a). By linking metacells to original cell type designations, we observe subpopulations of T cells, dendritic cells (DCs), and macrophages that are differentially abundant in lean and obese eAT (Fig. 3b), consistent with published literature[6,24,25]. Furthermore, metacells were reclassified by high-resolution subclusters to identify potential subpopulations of interest, such as regulatory T cells (T$_{regs}$), type 2 innate lymphoid cells (ILC2s), and tissue resident macrophages (TRMs) – abundant in the lean state; and LAMs, effector memory CD8$^+$ T cells (CD8$^+$ T$_{EM}$) and activated DCs – abundant in the obese state (Fig. 3c). By tracing cells back to each individual mouse, we were also able to perform pairwise comparisons of cell number for cell subtypes classically associated with lean AT (Fig. 3d) and obese AT (Fig. 3e). These data validate our cell isolation strategy and cell type identification by confirming, with high fidelity, many of the previously established changes associated with lean and obese AT.

**Obesity-associated T cell exhaustion persists after WL.** Adaptive T-lymphocytes are important regulators and drivers of inflammation in AT. We explored AT T cells by scSeq (Fig. 4a) and identified specific subsets of α/β-T cells by expression of

conventional T cell markers (*Cd3ε, Cd4, Cd8b1*), phenotype markers (*Foxp3, Cxcr3, Ccr7, Sell*)[26,27], and markers of cell cycling (*Stmn1, Pclaf*)[28,29] (Fig. 4b). A population of γ/δ-T cells was identified by expression of the common T cell delta chain (*Trdc*) and both natural killer (NK) and natural killer T cells (NKT) were identified by expression of killer lectin receptors *Klrb1b* and *Klrb1c* with and without expression of *Cd3ε*, respectively[30] (Supplementary Fig. 7). Th1 CD4$^+$ and all three CD8$^+$ T cells subsets were not elevated in WC but CD8+ T$_{EM}$ were increased in AT from WL mice (Fig. 4c). T$_{regs}$ expressing *Foxp3* trended towards being reduced in proportion by obesity and remain low following WL and WC. Using differential expression, we observed that obesity not only affected T$_{reg}$ number but also reduced the expression of *Il1rl1*, which codes for the IL-33 receptor, ST2 (Fig. 4d). Changes in other IL-33 responsive cell types, such as ILC2s and mast cells were also observed following obesity, failed to recover with WL, and were exacerbated by WC (Supplementary Fig. 8).

Because CD8$^+$ T cells have the capacity to clonally expand in response to antigen, we were interested in exploring if cycling T cells (marked by expression of *Pclaf, Stmn1*, and *Mki67*)[28] are precursors for AT memory T cell populations. Therefore, we utilized RNA velocity, which estimates cell state progression by comparing the frequency of spliced and unspliced mRNA sequences between cells. RNA velocity indicates cycling CD8$^+$ T cells are likely upstream of other effector and memory populations in AT, but also that bidirectional transition between effector memory and circulating memory can occur (Fig. 4e). Differential expression across diet groups also identified that cycling and T$_{EM}$ CD8$^+$ T cells express the activation/exhaustion signature gene *Pdcd1* (coding for PD-1). To further explore whether AT T cells contain an exhausted signature following obesity, we generated an exhaustion module containing multiple established gene expression features of T cell exhaustion: *Pdcd1, Tox, Entpd1, Tigit*, and *Lag3*[31,32]. We observed that T cells from AT of obesity, WL, and WC were all enriched for our exhaustion module and that CD8$^+$ T$_{EM}$ were most associated with an exhausted phenotype (Fig. 4f). This was further confirmed by a T cell exhaustion module based on protein expression of PD-1 (CD279) and TIGIT from our CITE-sequencing antibodies (Fig. 4g). T cell exhaustion is frequently studied in models of viral infection and in the tumor microenvironment, so we utilized ProjecTILs V1.0.0, a published scRNA-seq reference atlas[33], to further confirm our exhaustion profile. We observed that cells captured in our experiments projected more accurately onto an LCMV chronic infection atlas (Fig. 4h) than to a tumor-infiltrating lymphocyte (TIL) atlas (Supplementary Fig. 9). Concordant with our exhaustion module, more CD8$^+$ T cells aligned with the LCMV exhausted precursor cells in obese, WL, and WC groups than in the lean control group.

**Monocytes are abundant in AT and have modest functional changes that aren't restored with WL.** The AT immune cell

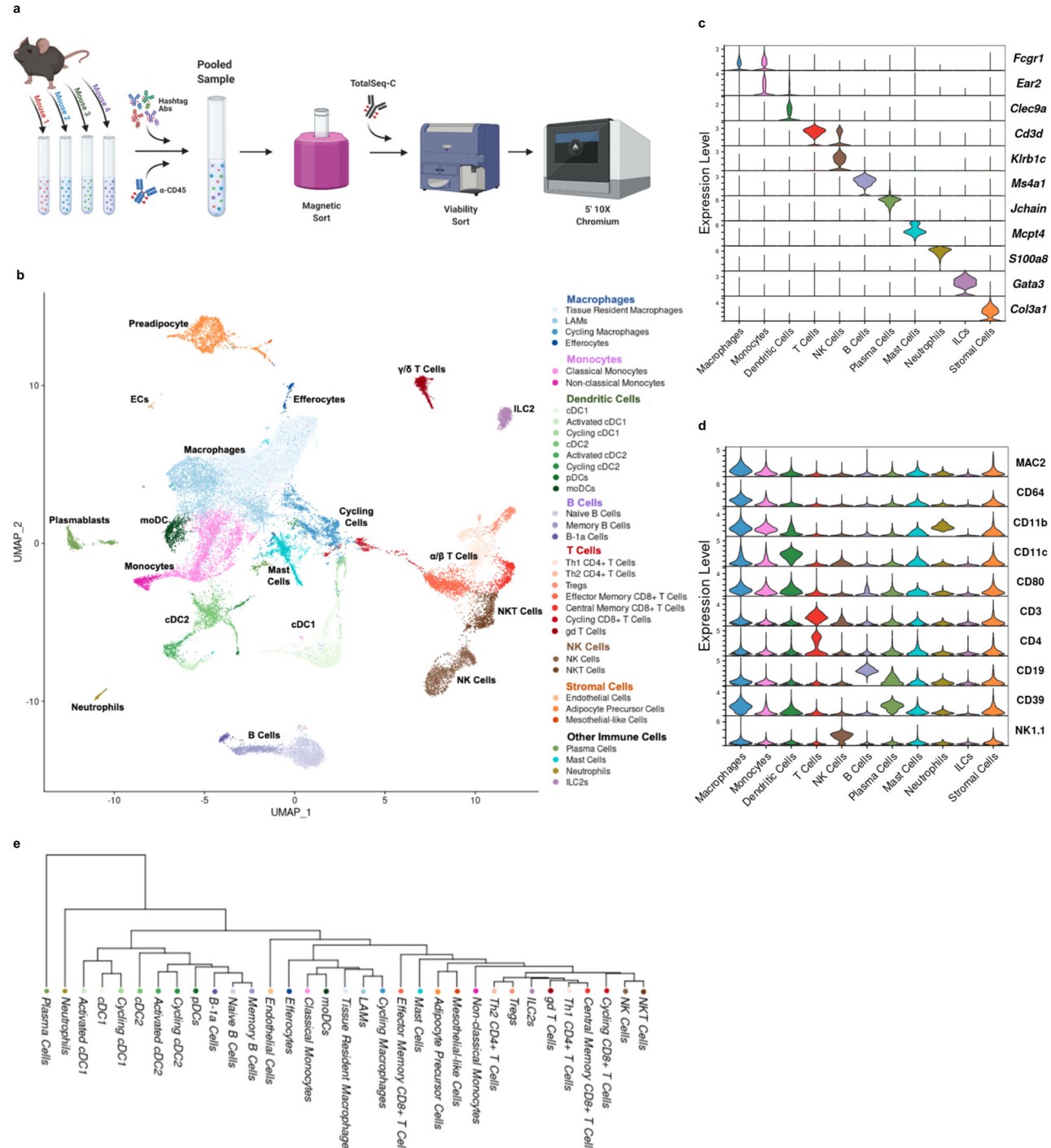

**Fig. 2 Adipose tissue immune cell populations observed by CITE-seq.** Schematic of CITE-seq approach using hashtag antibodies (Abs) and Total-SeqC antibodies. **b** Unbiased clustering of 33,322 single cells labeled broadly by cell type category and colored by high-resolution cell type identities via Uniform Manifold Approximation and Projection (UMAP). Populations include lipid-associated macrophages (LAMs), conventional dendritic cells (cDCs), plasmacytoid DCs (pDCs), monocyte-derived DCs (moDCs), T helper (Th) cells, T regulatory cells (Tregs), gamma-delta (gd) T cells, natural killer (NK) cells, and type 2 innate-like lymphoid cells (ILC2s). Selected markers of specific cell subsets based on **c** gene expression and **d** surface protein. **e** Phylogenetic tree of high-resolution cell type identities. Figure 2a was created with Biorender.com.

compartment contains a large population of both monocytes and DCs (Fig. 5a). Monocytes were subclustered into classical ($Ly6c2^+Ccr2^+Cx3cr1^+$) and non-classical ($Ly6c2^-Ccr2^-Cx3cr1^+$) subsets[34,35] (Fig. 5b). Additionally, we observed that many classical monocytes expressed *Fcgr1* and that *Ear2* and *Ace* were highly specific for non-classical monocytes in AT. We did not observe strong correlation between diet group and number of

either classical or non-classical monocytes (Fig. 5c). While monocyte recruitment to the adipose tissue is observed with obesity, population changes are time-dependent and often masked by large changes in the proportion of other cell types[18,19]. Upon further assessment of cytokine, chemokine, and other functional markers, there were few differences in non-classical monocytes, but we did observe an increase in genes associated

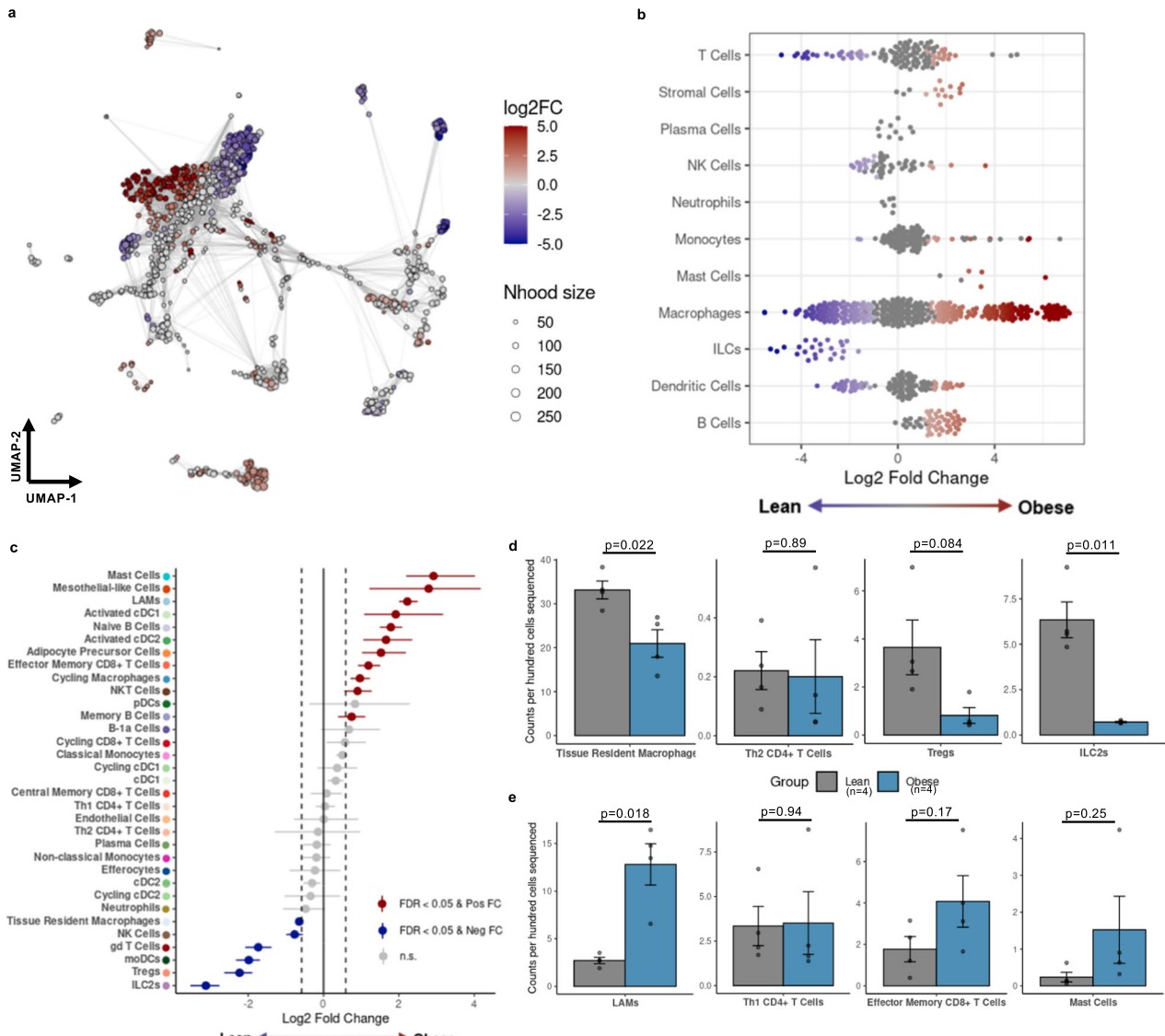

**Fig. 3 CITE-seq recapitulates obesity-associated immune cell changes in adipose tissue. a** Differential abundance by log fold change (FC) of metacells (index cells representing a neighborhood (Nhood) of cells connected by proximity in Uniform Manifold Approximation and Projection; UMAP) comparing cells from obese mice to lean mice. **b** Differential abundance of annotated cell types comparing cells from obese mice to lean mice. **c** Permutation testing of high-resolution clusters to calculate the proportional difference comparing cells from obese mice to lean mice by false discovery rate (FDR). Populations include lipid-associated macrophages (LAMs), conventional dendritic cells (cDCs), plasmacytoid DCs (pDCs), monocyte-derived DCs (moDCs), T helper (Th) cells, T regulatory cells (Tregs), gamma-delta (gd) T cells, natural killer (NK) cells, and type 2 innate-like lymphoid cells (ILCs). **d, e** Counts per hundred cells sequenced for lean-associated immune cell subsets and obesity-associated immune cell subsets in lean and obese mice (mean ± SEM; $n = 4$ mice; two-tailed $t$-test with indicated $p$ values). For diet groups, gray = lean, blue = obese.

with lipid handling (*Trem2*, *Cd36*, *Cd9*)[18], activation/adhesion (*Cd9*, *Cd81*, and *Cd63*)[36,37], and co-stimulation (*Cd86*, *Cd40*)[38], which were not reversed with weight loss in the classical monocyte subset (Fig. 5d). While *Cd86* gene expression increased following obesity, no change in *Cd80* mRNA or protein expression was observed due to diet within the classical monocyte subcluster.

**DCs shift towards an activated transcriptional signature with obesity and these signatures are retained with WL and WC.** DCs, a class of professional antigen-presenting cells (APCs), make up ~10–15% of captured AT immune cells. We classified DCs into two conventional DC (cDC) subsets, cDC1s (*Clec9a*⁺*Xcr1*⁺), and cDC2s (*Sirpa*⁺*Cd209*⁺)[39,40], and further refined subclustering using markers of DC activation (*Ccr7*)[41] and cell cycle induction (*Pclaf* and *Stmn1*)[29] (Fig. 5e). AT DCs were largely unchanged in relative

proportion by diet condition with the exception of monocyte-derived DCs (moDCs), which were enriched in lean AT (Fig. 5f). Activated subsets of DCs were elevated in obese AT and remained elevated upon WL and WC. Expression of *Ccr7* correlated positively with expression of *Fscn1* and *Mreg* and negatively with expression of cDC1 and cDC2 markers, indicating that activated DC subsets are mature (Fig. 5g). *Il15ra*, which codes for the receptor that trans-presents IL-15 to NK and T cells to support homeostatic proliferation[42,43], was also correlated with *Ccr7*. RNA velocity was used to determine whether activated DCs were precursors or downstream of other cDC subsets and suggested that activated cDCs mature from other local cDC subsets (Fig. 5h). Expression of *Ccr7* also negatively correlated with lactate dehydrogenase expression (*Ldha*), suggesting the potential for a shift in metabolite usage upon activation. Activated cDC1s, but not cDC2s, also expressed

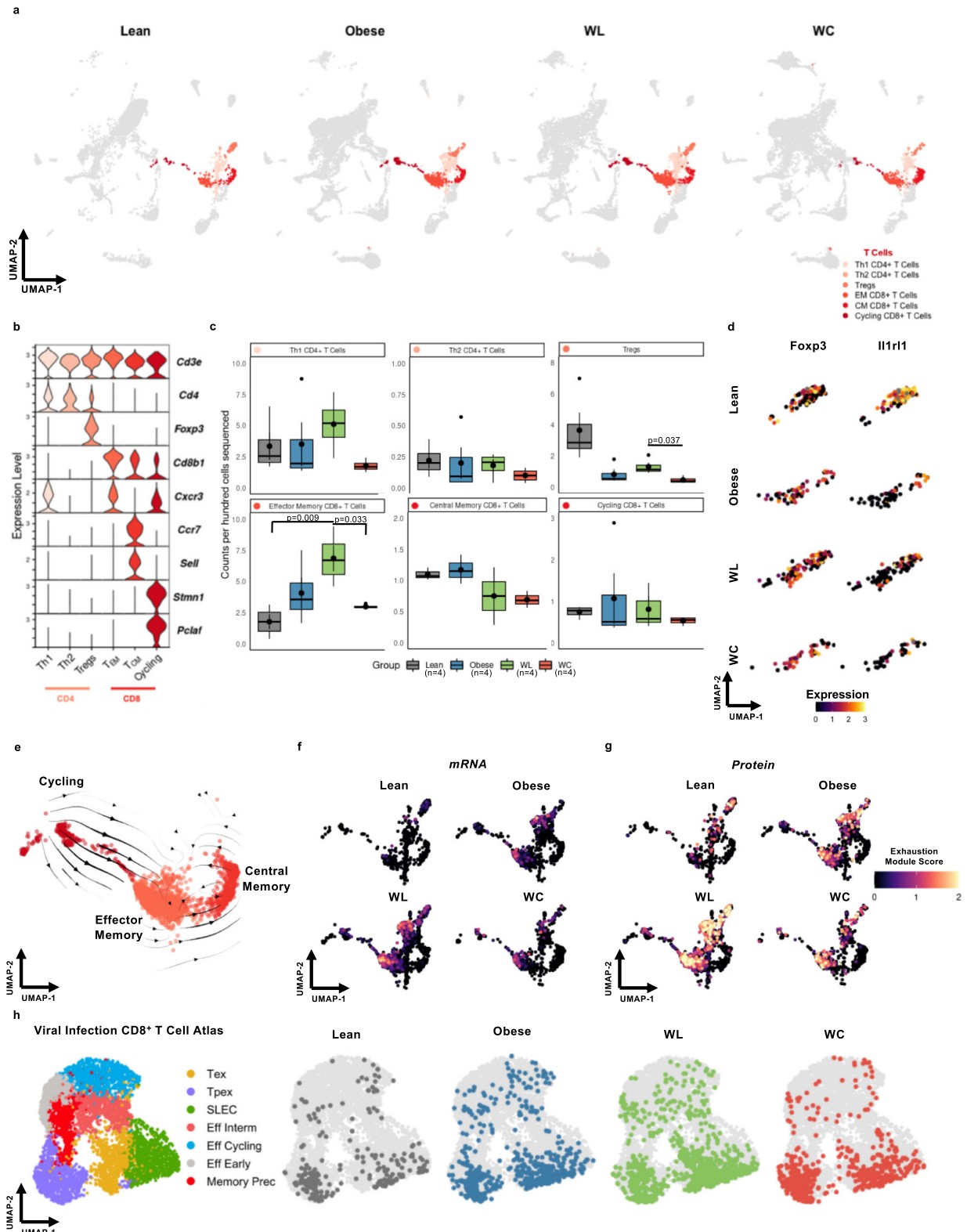

high levels of *Il12b* and may be primed to induce a local Th1 response within AT. The activated DC subset also expressed very high levels of key immunoregulatory ligands *Cd274*, *Pdcd1lg2*, and *Cd200* associated with coinhibitory T cell receptors (Fig. 5i). Taken together, these data suggest that DCs in AT shift towards an activated immunoregulatory status with onset of obesity and that these features are retained long-term through WL and WC.

**Macrophage populations are highly adaptable to change in dietary status**. Macrophages make up the largest proportion of immune cells in AT and are highly responsive to changes initiated by caloric excess. Dimensional reduction and clustering of macrophage subsets highlights the dramatic changes that occur during AT adaptation to HFD (Fig. 6a). Common markers of tissue macrophages, *Lyz2*, *Cst3*, *Adgre1* (coding for F4/80), *Cd68*, and

**Fig. 4 Adipose tissue T cells are retained and express markers of exhaustion in mice that have gained, lost, and regained weight. a** Uniform Manifold Approximation and Projection (UMAP) of T cell subclusters by diet group for lean, obese, weight loss (WL), and weight cycled (WC) mice. Populations include T helper (Th) cells, T regulatory cells (Tregs), and effector memory (EM) and central memory (CM) CD8+ T cells. **b** Expression of markers enriched in T cell subclusters. **c** Counts per hundred cells sequenced for α/β T cell subclusters (mean ± SEM; n = 4 mice). Box indicates interquartile range (25th–75th percentile) with 50th percentile indicated by solid line and mean indicated by large circle. Range of whiskers indicates largest and smallest values within 1.5 times the interquartile range and values outside of the range are indicated by small circles. **d** Expression of the $T_{reg}$ markers *Foxp3* and *Il1rl1* within the $T_{reg}$ subcluster across diet groups. **e** Embedding of RNA velocity displayed on the UMAP for CD8+ T cells. **f** CD8+ T cells colored by an exhaustion module containing the mRNA features *Pdcd1, Tox, Tigit, Lag3, and Entpd1.* **g** CD8+ T cells colored by an exhaustion module containing the CITE-seq features PD-1 (CD279) and TIGIT. **h** CD8+ T cells plotted onto a viral infection CD8+ T cell reference atlas using ProjecTILs. ProjectTILs populations CD8+ include terminally-exhausted (Tex), CD8+ precursor-exhausted (Tpex), short-lived effector cells (SLEC), effector interim (Eff Interim), and memory precursors (Memory Prec). For diet groups, gray = lean, blue = obese, green = WL, orange = WC. Each T cell subset indicated with a different shade of red. Pairwise two-tailed *t*-tests with Bonferroni correction for multiple comparisons were used to compare groups for cell counts; significant p values shown.

*Lgals3* (coding for MAC2) were highly expressed by all macrophage subclusters and robust protein expression for conventional AT macrophage markers CD64 and CD11b was observed (Fig. 6b). TRMs were enriched for expression of *Klf4, Cbr2, and Stab1*[44], while LAMs highly expressed various genes associated with lipid interactions (*Trem2, Cd9, Lpl*)[18], and cycling macrophages expressed cell cycle genes (*Stmn1, Pclaf*)[29] (Fig. 6c). In addition, we identified a subset of cycling macrophages and another small subset of macrophages that expressed very high levels of *Saa3 and Slpi*, which are defining features of efferocytes[45].

The changes in macrophage subclusters largely persist after weight gain, even following 9 weeks of WL or WC. TRMs decreased with obesity and even more with WL and WC (Fig. 6d). Importantly, while LAMs increase with obesity, they do not return to lean levels with WL and increase even more with WC; thus, levels are not directly correlated with AT mass. Cycling macrophages are increased with obesity, recovered with WL, but again worsen with WC and frequency of efferocytes was similar between diet groups.

**Alterations in macrophage phenotype remain unresolved with WL.** A great challenge in classification of AT macrophages is that conventional markers are often not exclusive or only poorly describe cell function. We observed that TRMs retained high expression of the M2-associated *Mrc1* gene (coding for the mannose receptor CD206). However, these same cells completely lost expression of the M2-associated *Cd163* gene with the onset of obesity (Fig. 6e). The persistence of this transcriptional change, which lasted for at least 9 weeks following WL and into subsequent WC, compelled us to further investigate if TRMs could be transitioning towards a LAM-like profile. Based on RNA velocity estimates, the majority of LAMs are likely derived from tissue infiltrating monocytes, as previously suggested[18], that acquire features of lipid handling prior to differentiation (Fig. 5d). However, RNA velocity also indicates that some TRMs are projected to become LAMs (Fig. 6f). MacSpectrum, a tool that uses macrophage differentiation (MDI) and polarization indexes (MPI) previously generated using in vitro systems[17], was utilized to further interrogate changes in tissue-resident and LAM phenotypes (Fig. 5g). We observed that obesity shifted both subpopulations of macrophages towards a more pro-inflammatory phenotype that was not recovered following WL, indicated by a higher MPI which signifies gene expression patterns associated with M1-like phenotypes. Specifically, WC LAMs appeared to be even more inflammatory compared to cells from other conditions, indicating that these cells may be an important target for subsequent studies seeking to improve outcomes of WL and weight regain.

## Discussion

WL improves obesity-associated insulin resistance in humans and mouse models; however, as highlighted here and by others[8,19,46,47], WL does not normalize AT immune populations

in murine models of WC. Previous research shows that AT immune cells increase during early stages of WL[46,48], presumably functioning as lipid handlers during lipolysis. Following WL in our model, male mice still retain many of the changes associated with prior obesity. We speculate that this obesity-associated immunophenotypic imprinting, which does not recover with WL, may ultimately be an underlying contributor to the detrimental impact of weight regain on metabolic health that is observed in both mice and humans. Supporting this, many inflammatory phenotypes are exacerbated in the WC group compared to the obese group. Indeed, given the frequency and risks associated with weight regain[7–13], understanding the cell types and/or mechanisms in AT that are not fully recovered following WL is critical.

In the T cell compartment, the greatest change in abundance occurs in $T_{regs}$, which decrease with obesity and do not rebound with WL. Interestingly, obese AT $T_{regs}$ have decreased expression of *Il1rl*, which codes for the receptor ST2, and *Il1rl1* expression remains low with WL and WC. This is consistent with *Gata3*-expressing ILC2s, which are reduced with obesity and produce IL33 – a type 2 cytokine that promotes glucose tolerance in mice and is the primary ligand for ST2[49]. AT $T_{reg}$ ST2 expression is regulated by *Pparg* in response to insulin in a HIF-1α and MED23 dependent manner, and impairments in this signaling reduce stimulated proliferation of $T_{regs}$[50]. The persistent loss of ST2 expression in AT $T_{regs}$ during WL and WC suggests a cell-intrinsic $T_{reg}$ imprinting induced by obesity that ultimately reduces their long-term maintenance for protection against glucose intolerance during future weight regain. Interestingly, mast cells are increased with WC, but also have reduced *Il1lr* expression and increased expression of lipid-associated genes, such as *Trem2* and *Fabp5*. While the role of mast cells in AT is unclear due to lack of specific knockout models, this suggests that the demand to handle excess lipid in obese AT occurs concomitantly with reductions in availability and response to type 2 cytokines, such as IL-33, and that these changes are not resolved with WL.

CD8+ T cells are elevated in obesity and WC and are most enriched after WL. Moreover, obesity increases expression of exhaustion-associated genes in CD8+ T cells that is not normalized by WL. T cell exhaustion has been recently noted in human and mouse AT CD8+ T cells, which were shown to have impaired stimulation and increased markers of T cell exhaustion following obesity[51–53]. This transcriptional phenotype was further validated in our model by comparing T cells identified in our study to previously published exhausted T cell atlases generated in models of viral infection and cancer[33]. We also previously reported that AT T cell clonality is increased during obesity and that the T cell repertoire likely responds to positively charged, non-polar antigens[54]. It is plausible that APCs accumulate and present lipid-adducted proteins during lipid clearance following weight gain but also WL. Thus, chronic CD8+ T cell stimulation

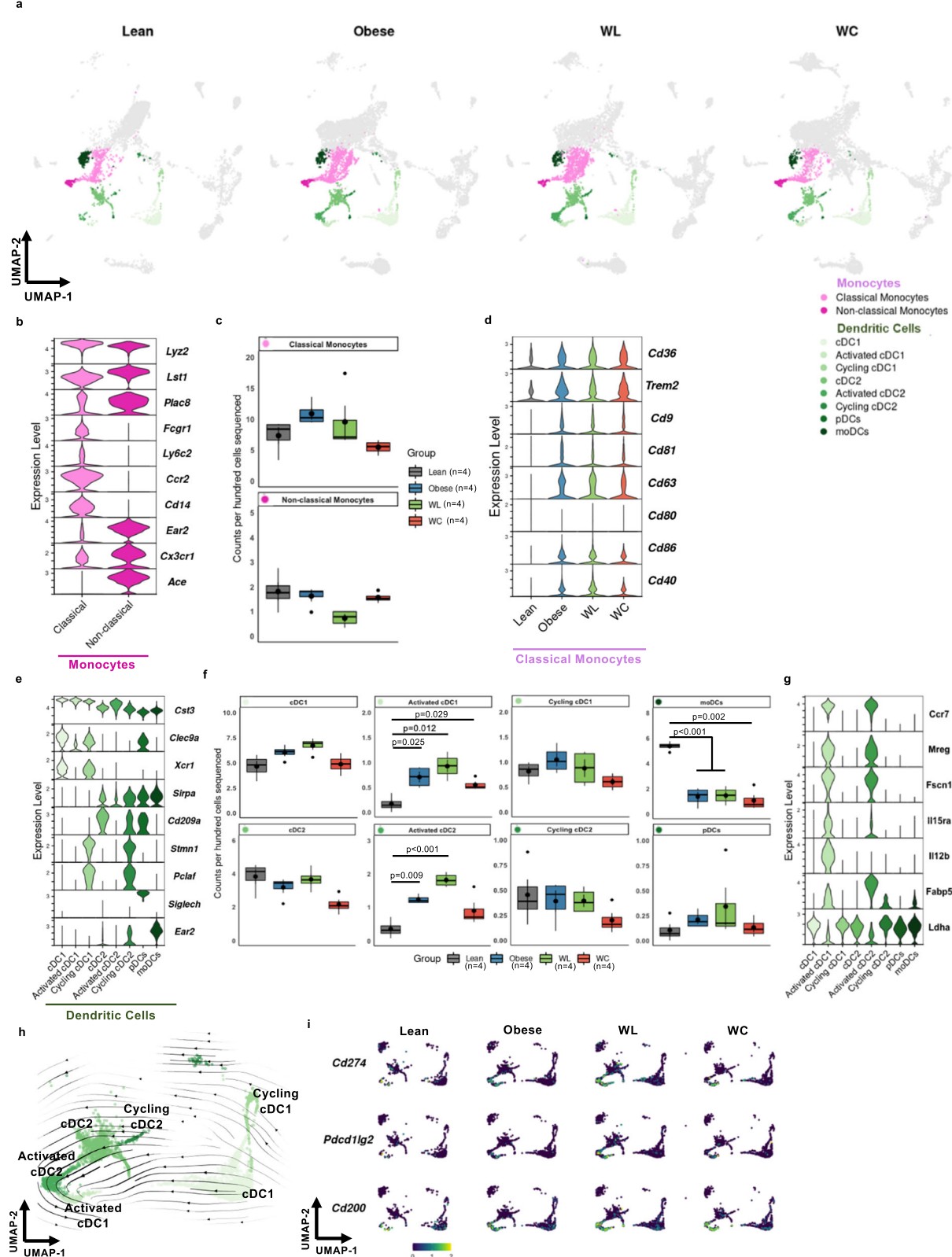

via antigen presentation may drive T cell exhaustion and warrants further investigation. In support of this, AT DCs shift to a mature, activated state with the onset of obesity and persist during WL and WC. This activated status is characterized by increased expression of *Ccr7, Mreg,* and *Fscn1*, which have all been previously reported as critical features of immunoregulatory DCs that are enriched in non-small-cell lung cancer and uptake

of anti-tumor antigens[55]. Furthermore, expression of the immunoregulatory proteins *Cd274, Pdcd1lg2,* and *Cd200* on activated AT DCs suggests these cells may be important regulators of T cell function and exhaustion in AT.

Macrophages are the most abundant immune cell type in our analysis. TRMs decrease with obesity and do not recover with WL. We also noticed a remarkable change in transcriptomic

**Fig. 5 Monocytes and dendritic cells (DCs) are abundant in adipose tissue and DCs shift towards a mature, activated status in mice that have gained, lost, and regained weight. a** Uniform Manifold Approximation and Projection (UMAP) of highlighted monocyte and DC subclusters by diet group for lean, obese, weight loss (WL), and weight cycled (WC) mice. Populations include conventional dendritic cells (cDCs), plasmacytoid DCs (pDCs), and monocyte-derived DCs (moDCs). **b** Expression of genes associated with monocyte subclusters. **c** Monocyte counts per 100 cells sequenced by diet group (mean ± SEM; $n = 4$ mice; n.s.d.). **d** Lipid-handling, activation, adhesion, and co-stimulation genes by diet group for classical monocytes. **e** Expression of genes enriched in DC subsets. **f** DC counts per 100 cells sequenced by diet group (mean ± SEM; $n = 4$ mice). **g** Expression of genes associated with DC activation. **h** Embedding of RNA velocity displayed on the UMAP for conventional DC subsets. **i** Expression of immunoregulatory ligands *Cd274*, *Pdcd1lg2*, and *Cd200* in DCs by diet group. For diet groups, gray = lean, blue = obese, green = WL, orange = WC. Each monocyte subset indicated with a different shade of pink and each DC subset with different shade of green. Pairwise two-tailed T-tests with Bonferroni correction for multiple comparisons were used to compare groups against the Lean reference group for cell counts with significant *p* values shown. For panels **c** and **f**, box indicates interquartile range (25th–75th percentile) with 50th percentile indicated by solid line and mean indicated by large circle. Range of whiskers indicates largest and smallest values within 1.5 times the interquartile range and values outside of the range are indicated by small circles.

profile of this subcluster with obesity. Expression of *Mrc1*, coding for the M2-like macrophage marker CD206, is maintained during obesity, WL, and WC. However, expression of *Cd163*, another M2-like marker, is lost with the onset of obesity and the loss persists during WL and WC. These findings support divergence from the classical M1-M2 terminology, particularly in disease-associated microenvironments like obese AT, in favor of functional phenotyping or high-dimensional cell phenotyping[56]. Moreover, LAMs increase with obesity and only partly resolve with WL, supporting the notion that WL alone is insufficient to correct the AT immune landscape. Upon weight regain, LAMs and cycling macrophages tend to increase compared to obese animals. This shift in proportions of TRMs and LAMs, and increased lipid handling genes in the classical monocyte population, suggests a critical role for lipid regulation following weight gain, however, the role of macrophage lipid handling is not fully understood[57,58]. LAM depletion via *Trem2* knockout has been reported to worsen insulin resistance during HFD feeding, suggesting macrophages may help buffer lipid overload[18]. Additionally, loss of TREM2 via global knockout or anti-TREM2 neutralizing antibodies improves T cell responses, similar to anti-PD-1 immunotherapy, and ultimately reduced tumor size in a macrophage-dependent manner[59]. Thus, lipid handling may be delicately linked to antigen presentation and T cell exhaustion in both metabolic disease and cancer. Interestingly, the tetraspanins CD9, CD63, and CD81 which were increased by gene expression in classical monocytes with WC have been suggested to play a role in multinucleate giant cell formation[60]. Giant multinucleated cells have been found in obese adipose tissue and contribute to the clearance of dead adipocytes[61,62]. However, in our studies, these large cells were likely filtered out during cell isolation, and thus it is not known if they change with WC.

In summary, we identified obesity-associated immune imprinting of multiple distinct cell subsets that likely contribute to WC-accelerated metabolic disease in our mouse model (Fig. 7), and potentially in humans. Unfortunately, the role of many of these cell types and functions in regulating adipose homeostasis during weight gain are not well understood, and even less is known in WL and WC. However, our results suggest critical areas of interest for future studies. Increases in activated DCs, LAMs, and exhausted T cells suggest that antigen presentation is a critical point of regulation, or dysregulation, in obese AT. Future studies could greatly improve our understanding of AT antigen presentation by utilizing single cell T cell receptor sequencing to uncover clonal responses and antigens associated with metabolic disease. Macrophage lipid handling also appears to be important for AT regulation; however, whether this process reaches capacity or is dysregulated in obesity is not known. Moreover, dysregulation of macrophage lipid handling and the lipid scavenger receptor, TREM2, have been implicated as key features in numerous diseases, such as Alzheimer's, cancer, and infection –

expanding the importance of our findings to additional immunological diseases. Finally, the loss of type 2 immune subsets, IL-33 responsiveness, and changes in the TRM transcriptional phenotype along with a concomitant increase in M1-like cells, support an increase in inflammation, which has been previously linked to insulin resistance. Taken together, these findings support continued investigation into regulation of AT immune cells during obesity, WL, and WC in both murine models and humans. Understanding which processes are critical for adipose homeostasis may help suggest pharmacological interventions that can be used during WL or weight regain to mitigate metabolic disease after WC. Additionally, studies utilizing other models of WL, such as exercise, bariatric surgery, and pharmacological WL, could greatly improve our understanding of immunometabolism and will help to identify advantageous immunotherapeutic targets.

**Considerations.** Several aspects of this study require additional consideration. First, only male mice were included in the immunological studies, thus we were unable to determine whether the observed changes in the adipose immune compartment linked with WL and WC manifest similarly in females. It is well established that female rodents are less susceptible to diet-induced obesity than males and display a different inflammatory phenotype in adipose tissue than males[63]. Moreover, females have greater adipose and systemic insulin sensitivity than males for a given body mass[64–67], and the response to caloric restriction and subsequent hyperphagia following ad libitum food access is lower in female mice[68]. Thus, as expected, female mice in this study gained less weight than males on HFD. WC females did not markedly worsen glucose tolerance, however, their initial bout of weight gain following 9 weeks of HFD was minimal. We hypothesize that if female mice initiated the study at an older age when body weight gain is steeper on HFD, the subsequent WC phenotype (i.e., augmented glucose intolerance) would manifest. It is also likely that greater weight fluctuations are required to worsen metabolic function in females. Nonetheless, it is probable that at least in this model, WC females would induce distinct immune remodeling compared with males. Future studies could use different models (such as starting diet at an older age when body weight gain is steeper or ovariectomized animals) to assess the differences in male and female responses to WC and the degree of weight variability required to observe metabolic differences.

Importantly, the immunometabolic differences in obese and WC mice occur independent of body weight, total fat mass, and lean mass. There was a small, but significant increase in eAT mass and the proportion eAT mass to total body mass in the WC compared to the obese group, and a small but significant increase in lean mass in the WL compared to the lean group. However, with differences <1 g between groups, they are unlikely to drive the results. This work represents only one cycle of WL and regain

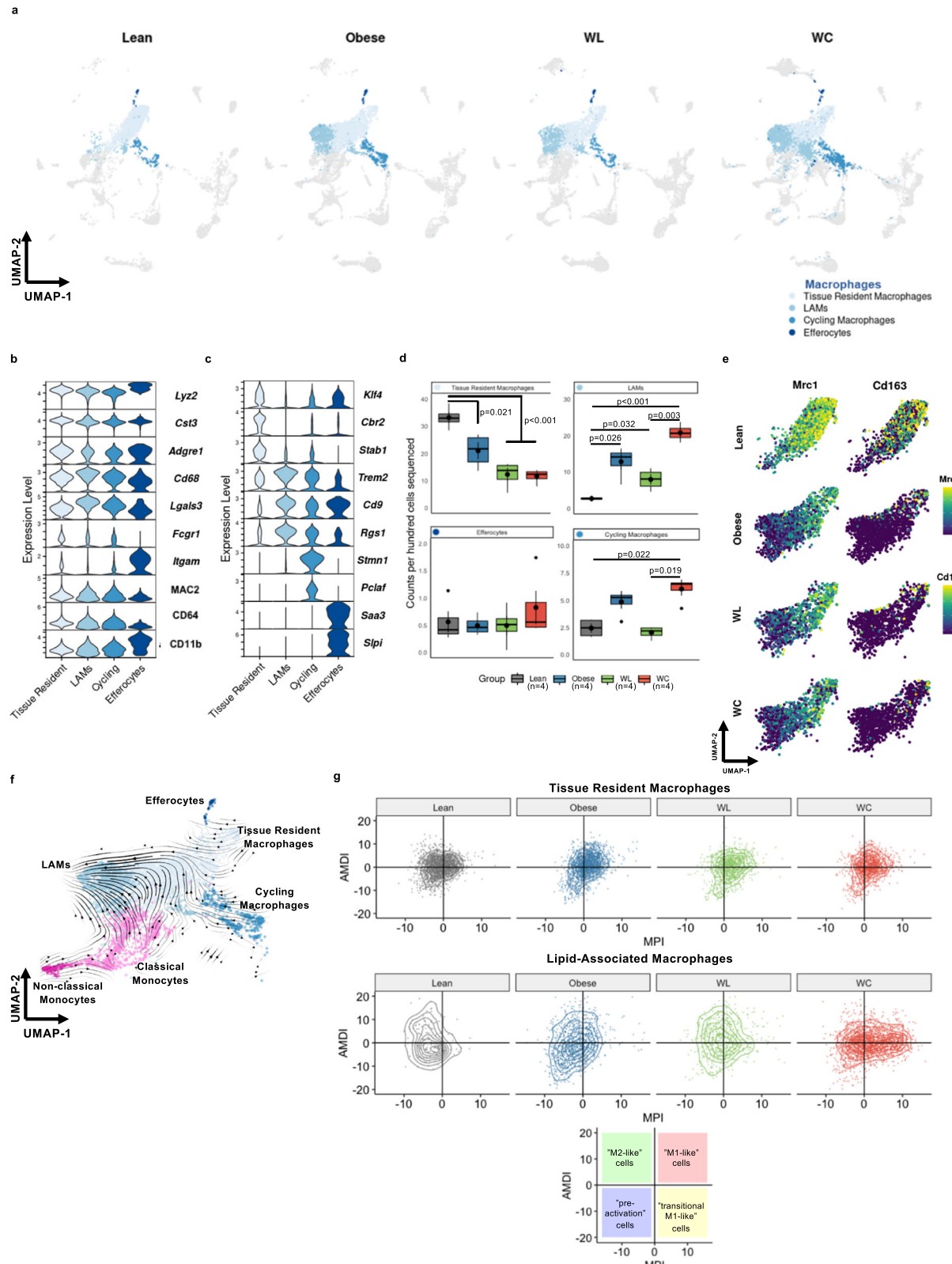

and utilizes a switch from primarily high fat to low fat feeding, which likely differs from human WC. Others have also shown that multiple weight change cycles worsen glucose tolerance[69]. Moreover, other WC models have demonstrated that WC reduces adiponectin and CRTP3[70,71], downregulates clock genes[72], increases fat regain due to the loss of lean mass with WL which increases appetite, reduces energy expenditure, and reduces adaptive thermogenesis[69,73,74]. The immune profile of WC mice was not evaluated in any of these models, and it is likely that models with greater weight gain across multiple cycles would show even more immunological difference. As diet composition can affect adipose biology[48], models using caloric restriction or

**Fig. 6 Diet-induced obesity causes persistent changes in adipose tissue macrophages, even after weight loss and regain. a** Uniform Manifold Approximation and Projection (UMAP) of macrophage subclusters plotted by diet groups for lean, obese, weight loss (WL), and weight cycled (WC) mice. Populations include lipid-associated macrophages (LAMs). **b** Expression levels for genes (*Lyz2, Cst3, Adgre1, Cd68, Lgals3, and Itgam*) and proteins (CITE-seq; MAC2, CD64, and CD11b) associated with macrophages. **c** Expression of genes that associate with macrophage subclusters. **d** Counts per hundred cells sequenced for macrophage subclusters (mean ± SEM; $n = 4$ mice). Box indicates interquartile range (25th–75th percentile) with 50th percentile indicated by solid line and mean indicated by large circle. Range of whiskers indicates largest and smallest values within 1.5 times the interquartile range and values outside of the range are indicated by small circles. **e** UMAP visualization of *Mrc1* and *Cd163* expression in tissue resident macrophages (TRMs) by diet group. **f** Embedding of RNA velocity displayed on the UMAP for macrophage and monocyte subsets. **g** TRMs and LAMs plotted on based on the macrophage polarization index (MPI) and the activation-induced macrophage differentiation index (AMDI) calculated using MacSpectrum. A shift to the right indicates a more M1-like phenotype. For diet groups, gray = lean, blue = obese, green = WL, orange = WC. Each macrophage subset indicated with a different shade of blue. Pairwise two-tailed *t*-tests with Bonferroni correction for multiple comparisons were used to compare groups for cell counts; significant *p* values shown.

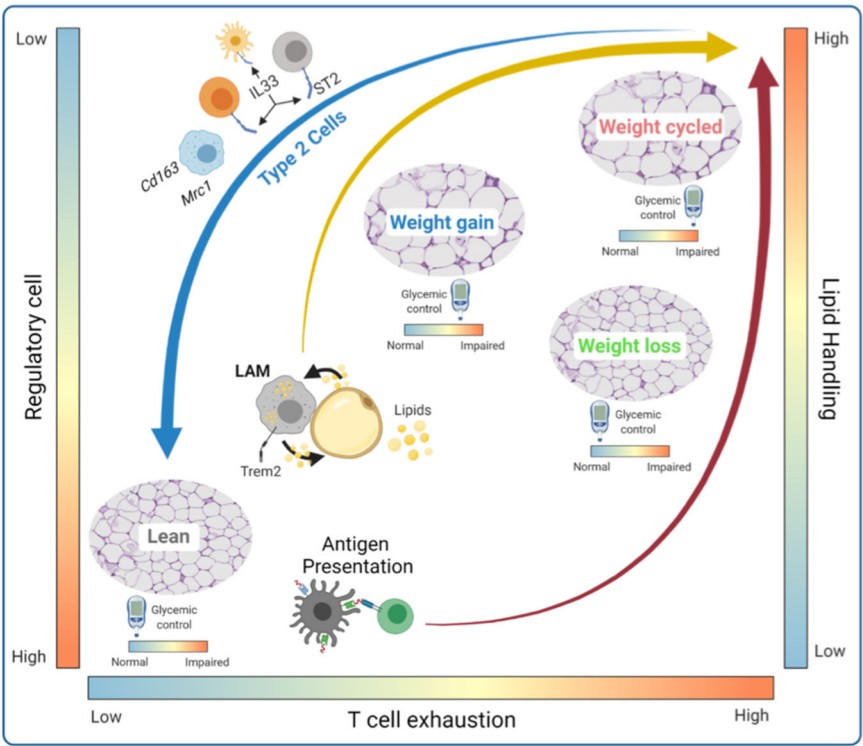

**Fig. 7 Adipose tissue immune remodeling during WL and WC.** Body weight gain results in a transition from abundant type 2 immune cells cells (type 2 innate-like lymphoid cells, regulatory T cells, and tissue resident macrophages) present in lean adipose tissue towards an accumulation of immune cells/phenotypes associated with lipid handling (lipid-associated macrophages, LAMs) and antigen presentation. WL does not restore the antigen presentation phenotype and is associated with T cell exhaustion. T cell exhaustion and LAMs are further amplified by weight regain. There is a disconnect between systemic glucose homeostasis and immune remodeling in adipose tissue, such that WL corrects obesity-induced glucose intolerance, but does not resolve the altered immune cell composition caused by diet-induced obesity.

altered diet composition, as well as exercise, pharmacological, bariatric surgery, or environmental temperature would greatly improve our understanding of WC-associated disease.

The development of scSeq has been especially important in the field of Immunometabolism, as the dichotomy of immune cells as strictly type 1 or type 2 is inadequate and measuring changes in general cell populations mask inflammatory changes as well as other functional activity such as lipid handling that appears within cell type subclusters. We found 11 primary clusters and 33 unique subclusters, validated with CITE-seq antibodies for identifying cell surface markers. However, we did not capture all known AT immune populations. For example, we did not identify eosinophils, ILC1/3, B1-a, and B2 cells, or iNKT cells in our dataset, which may be due to isolation methodology, inherent cell properties, and sequencing parameters. Eosinophils contain many RNAses for pathogen defense that are released during cell lysis. Moreover, it's plausible that NKT and iNKT cells, for example, have differential cell surface expression that is not easily captured transcriptionally. Genes associated with transcription factors and cytokines were lowly expressed or dropped out in our data set, likely limiting our ability to identify cell types specifically distinguished by these markers. Use of stimulation conditions, increased sequencing depth, or CITE-seq targeted to intracellular epitopes[75], could further improve identification of addition cell states in future experiments. Additionally, the use of a much more expansive CITE-seq panel would greatly improve clustering and cell type annotation in future experiments.

Finally, our analysis was largely focused on the immune cell compartment of adipose tissue. However, we identified 1,836 cells that were annotated as stromal cells despite magnetically sorting for CD45. Doublet detection using DoubletFinder indicated approximately 10.2% of these cells may be heterotypic doublets

(compared to only 0.28% for all other clusters combined). Differential gene expression identified genes associated with macrophages, mast cells, and monocytes in stromal cells labeled as doublets compared to those labeled as singlets. Therefore, studies focused on adipocyte progenitors[76] or mature adipocytes[77–79] (captured via single nuclei sequencing) may provide more insight into the relationship between stromal and immune cells and could be expanded to include cells from WC mice.

**Open access interactive portal.** Importantly, we believe the immunophenotyping conducted using CITE-seq in our models of obesity, WL, and WC has broad applicability, which can be hypothesis-generating for models of cancer, infection, or metabolic disease in other tissues. Moreover, our dataset is a resource for identifying targets for focused investigations in clinical human AT samples. To facilitate discovery and to broaden accessibility of this data, we have created an open-access online interactive portal called MAIseq (Murine Adipose Immune sequencing) with our preprocessed data for the research community using modified source code from ShinyCell[80] at https://hastylab.shinyapps.io/MAIseq/. Users can utilize a variety of built-in visualization approaches that span beyond what we could report in this manuscript. For instance, we also identified γ/δ T cells, NK and NKT cells (Supplementary Fig. 7), multiple B cell subsets and plasma cells (Supplementary Fig. 10), and even some CD45+ stromal cells that we did not investigate here. Users can explore our cell cluster annotations, identify potential genes of interest, plot gene expression by diet groups, clusters, or subclusters, and plot expression of surface markers from CITE-seq. We provide clear instructions for use and options to download figures and tables in a variety of formats. Collectively, our data provides critical groundwork for understanding the causes of WC-accelerated metabolic disease.

## Methods

**Mice housing and mouse model.** Male C57BL/6 J mice were purchased from Jackson Labs at 7 weeks of age. At 8 weeks of age, mice were placed on 9 week cycles of 60% HFD (Research Diets #D12492) or 10% LFD (Research Diets #D12450B) for a total of 27 weeks previously reported[14] and shown in Fig. 1a. Additionally, a small cohort of female C57BL/6 J mice purchased from Jackson Labs at 7 weeks of age were weight cycled following the same design as for lean, obese, and weight cycled male mice. Body weight and food intake were recorded weekly. The room was maintained at a controlled temperature (~23 °C) and humidity (~30%) and 12 h light-dark cycles. All studies were approved by the Institutional Animal Care and Usage Committee of Vanderbilt University.

**Body composition and glucose tolerance.** At 26 weeks, body composition was measured in the Vanderbilt University Mouse Metabolic Phenotyping Center via nuclear magnetic resonance (Bruker Minispec). Lean body mass, body fat, and free body fluid were recorded. The following day, ipGTT was performed. Briefly, baseline fasting blood samples were obtained by cutting off the tip of the tail under isoflurane anesthesia. Glucose was injected intraperitoneally at 1.5 g/kg of lean mass and subsequent blood samples were collected at 15, 30, 45, 60, 90, 120, and 150 min post injection. Blood glucose was measured using a Contour Next EZ Blood Glucose Monitoring System and test strips (Bayer). Differences in glucose tolerance over time were assessed using two-way ANOVAs and differences in body composition and glucose tolerance test AUC were assessed using two-tailed T tests with Bonferroni corrections for multiple comparisons. Differences in tissue mass were assessed using two-tailed T tests with Bonferroni corrections for multiple comparisons. Statistical significance was set to adjusted $p < 0.05$.

**Immunofluorescence microscopy.** Tissues were fixed in 1% paraformaldehyde for 1 h and stored in 70% EtOH overnight before paraffin embedding by the Vanderbilt Tissue Pathology Shared Resource core. Tissues were sectioned at 5 μm and allowed to dry overnight. Slides were deparaffinized in xylene and dehydrated through an EtOH gradient, then immunolabeled with Rabbit anti-Perilipin-1 (Abcam #9349T; Clone D1D8) at 1:200 overnight at 4 °C. After washing, Goat anti-Rabbit IgG conjugated to AF647 (Abcam #ab150079) was applied for 2 h at room temperature prior to washing and coverslipping with Prolong Gold (Invitrogen #P36931) containing DAPI. Slides were imaged using a 20X objective on a Leica

DMI8 widefield microscope and captured with a Leica DFC9000GT camera. Image tiles were taken across the entirety of each section and stitched using the LAS X software suite. Merged images were processed in ImageJ and adipocytes were counted using an in-house macro. Briefly, the AF647 channel was processed by enhancing contrast and applying a gaussian blur with a sigma = 5. Background subtraction was performed using a rolling basis of 30. Auto thresholding using the "Triangle dark" setting was used to preserve adipocyte borders. The images were then skeletonized and then Image J's analyze particles function was used with a size threshold range of 300–315,000, circularity of 0.4–1. Only cells with a measured diameter >10 μ were included in our analysis.

**Stromal vascular fraction isolation and cell sorting.** Mice were euthanized by isoflurane overdose and cervical dislocation and perfused with 20 ml PBS through the left ventricle. eAT pads were collected, minced, and digested in 6 ml of 2-mg/mL type II collagenase (Worthington # LS004177) for 30 min at 37 °C. Digested eAT was then vortexed, filtered through 100 μm filters, lysed with ACK buffer, and filtered through 35 μm filters as previously described[81].

The AT stromal vascular fraction was prepped with anti-mouse Fc Block (BD Biosciences) at 1:200. Cells from each mouse were labeled with unique hashtag antibodies (1:200) (Biolegend TotalSeq-C) and anti-CD45 microbeads (10 μL/sample) (Miltenyi # 130-052-301). Biological replicates were pooled and sorted on a Miltenyi AutoMACs using the "possel_s" option. CD45+ cells were then labeled for CITE-sequencing using TotalSeq-C antibodies (Biolegend) for cell surface markers to identify major cell types. All immunolabeling was completed at a 1:200 dilution for 20 min at 4 °C in the dark. More information regarding specific samples and antibody manufacturer/catalog numbers can be found in Supplementary Table 1. Cells were stained with 0.25 μg/mL DAPI for FACS sorting and DAPI⁻ viable cells were collected for downstream processing and sequencing.

**Single cell RNA-sequencing.** All samples were submitted and processed for sequencing on the same day to minimize batch effects. Sample preparation was conducted using the 5′ assay for the 10X Chromium platform (10X Genomics) targeting 20,000 cells per diet group (~5000 cells per biological replicate). In all, 50,000 reads per cell were targeted for PE-150 sequencing on an Illumina Nova-Seq6000. Sample processing and sequencing were completed within the same run in the VANderbilt Technologies for Advanced GEnomics core (VANTAGE).

**Data processing.** FastQ files obtained from sequencing were processed using CellRanger V3 (10X Genomics) with feature barcoding. Outputs from CellRanger were further processed using Velocyto V0.17[82] for downstream RNA velocity analysis. The R package SoupX V1.5.2[83] was used to remove ambient contaminating RNA. The Seurat V4 R package[84] was used for quality control, data set integration, clustering, cell type annotation, differential expression, and visualization. The Python package scVelo V0.2.3[85] was used for modeling and visualizing RNA velocity through R using Reticulate V1.22. Strict quality control parameters were utilized post sequencing to ensure that only viable cells which could be traced back to each individual mouse were used for further analysis. For quality control of sequenced cells, only cells that had at least 200 gene features, at least 500 total measured RNA sequences, and <5% mitochondrial RNA content were retained. Furthermore, only cells designated as singlets after hashtag demultiplexing (i.e. cells containing at least one hashtag, but not more than one) were used for visualization and analysis. DoubletFinder V3[86] was used to detect heterotypic doublets and further confirm singlets. RPCA-based integrated was used to integrate the four sequencing datasets. For more detailed processing information, we have provided the minimal necessary data and three vignettes containing code for preprocessing, data integration, and cell type annotation at: https://github.com/HastyLab/Multiomics-WeightCycling-Vignettes.

**Annotating clusters.** Cluster annotation was performed following dimensional reduction using the integrated data assay. First, broad clusters were identified using low resolution (0.4) and annotated by common cell type markers. Each identified broad cluster, referred to as "Cell Types", were further subclustered and subclusters were annotated by comparing differentially expressed markers to published literature. Cluster annotations were further supported using SingleR V1.6.1[87] with the Immgen[88] and MouseRNAseq[89] databases from the celldex package V1.2.0[87].

**Downstream sequencing analysis.** All downstream analyses comparing cells across diet groups, individual biological replicates, or clusters were performed using the normalized RNA assay. Differential expression was performed using the Wilcoxon Ranked-Sums test in Seurat V4 for single cells. For pseudobulk analysis on our web-based tool, the likelihood ratio test in DESeq2 V1.32.0 was used. Differential abundance testing was conducted using the MiloR package V1.0.0[90] in R. Metaclusters identified by MiloR were further tested with permutation testing using the scProportionTest R package V0.0.9000 (https://github.com/rpolicastro/scProportionTest). For exploring macrophage phenotype, the R application MacSpectrum V1.0.1[17] was used with provided index information. RNA velocity was estimated using Veloctyo[82] and visualized using scVelo[85].

**Reporting summary**. Further information on research design is available in the Nature Research Reporting Summary linked to this article.

## Data availability

The in vivo data generated in this study are provided in the Source Data file. Raw sequencing files, processed data matrices, UMAP and PCA embeddings, cell metadata, and a fully integrated Seurat v4 R data object are available via the NCBI GEO with the primary accession code GSE182233.

## Code availability

Example code to reproduce our processed and integrated Seurat v4 object is available at: https://github.com/HastyLab/Multiomics-WeightCycling-Vignettes[91]. This GitHub repository contains information for (1) Preprocessing and Quality Control, (2) Data Integration, and (3) Cell Type Annotation. All code necessary to reproduce the figures within this manuscript are available at via GitHub at: https://github.com/HastyLab/Multiomics-WeightCycling-Figures[92]. Code for generating our ShinyCell-based web application is available upon request, but we highly recommend installation of ShinyCell directly through the author's GitHub page.

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

## Acknowledgements

This project was funded by a Veterans Affairs Merit Award 5I01BX002195 and an AHA Innovation Award (19IPLOI4760376) to A.H.H.; M.A.C. is funded by an NIH F31 Predoctoral Fellowship (1F31DK123881), H.L.C. is funded by an AHA Post-doctoral Fellowship (20POST35120547), and N.C.W. is funded by an AHA Post-doctoral Fellowship (21POST834990). M.A.C., H.L.C., and N.C.W. were all previously supported by the Molecular Endocrinology Training Program (T32 DK07563). The Translational Pathology Shared Resource used for tissue preparation is supported by NCI/NIH Cancer Center Support Grant 5P30 CA68485-19. FACS sorting was performed in the Vanderbilt Flow Cytometry Shared Resource which is supported by the Vanderbilt Ingram Cancer Center (P30CA068485) and the Vanderbilt Digestive Disease Research Center (DK058404). Single-cell preparation and sequencing were performed in the Vanderbilt Technologies for Advanced Genomics (VANTAGE) core laboratory.

## Author contributions

M.A.C. and H.L.C. completed the mouse work and cell isolation and processing for these experiments. M.A.C. performed data analysis and generated the interactive web portal. H.L.C. and M.A.C. drafted the manuscript. M.A.C. and N.C.W. produced the experimental design and summary figures using BioRender.com. A.H.H. conceptualized the study, provided funding, and is the guarantor for this work. All authors contributed to experimental design, data interpretation, website testing, and manuscript revisions.

## Competing interests

The authors declare no competing interests.
