## [Peer Review File · Nature Communications]

Reviewer comments, initial review

Reviewer #1 (Remarks to the Author):

The study describes the dynamics of adipose tissue immune cell populations in a mouse model of weight cycling (WC). There is an epidemiological rationale, which relies on the fact that in human obesity up to 60% of patients undergoing body mass reduction, will eventually present body mass regain; there is also a molecular/cellular rationale, which relies on the fact that adipose tissue inflammation provides an important mechanistic basis for insulin resistance and also for a number of other obesity comorbidities.

Here, authors employed CITE-seq to explore the putative changes in immune cell populations in the adipose tissue in body mass gain, body mass reduction and WC. They show that cellular markers of antigen presentation, T-cell exhaustion, lipid handling and inflammation persist after body mass reduction and worsen with body mass regain.

In general, this is a fine and timely study that provides methodological advance in the way we can study adipose tissue inflammation; it also provides huge amount of data that can be further explored by the group and also by others, in order to provide an in-depth analysis of the adipose tissue inflammation in obesity. Finally, and most importantly, the study provides important advance in the understanding of adipose tissue inflammation in obesity, showing that WC can worsen the immune phenotype of the adipose tissue infiltrates, which directly impact on the worsening of the metabolic phenotype.

Major issues

Authors should provide an expanded description of the CITE-seq method.

In page 5, authors say that a total of 33,322 cells met quality control. What was the number of cells per group? Was there any statistical difference in the number of cells obtained per group?

The WC model is interesting; however, it reflects the consequences of one cycle, only. In human obesity, patients frequently undergo several cycles of body mass reduction and regain. This should be acknowledged in the Discussion.

There is no description of the method and purpose of performing partition-based graph abstraction (PAGA). This should be included.

Monocyte recruitment to adipose tissue during the development of obesity is an important issue in this field. In their models, authors found no differences in the monocyte phenotypes; however, this seems odd. Authors could go deeper into the investigation of monocyte phenotypes looking into chemokine markers.

Minor issues

There is a typo on subsection title Body composition and glucose tolerance (page 15).

In page 11, the sentence – We postulate that this obesity-associate immunophenotypic imprinting... - is speculative as the study has neither evaluated cardiovascular outcomes nor the relation of the adipose tissue immune cells with putative cardiovascular abnormalities.

Reviewer #2 (Remarks to the Author):

In this study, Cottam et al successfully developed a mouse model representing weight loss-accelerated metabolic disease then performed CITEseq analysis on 33,322 immune cells in the adipose tissues from a total of 16 mice assigned into 4 diet groups. Major findings from these analyses include 1) Obesity-associated T cell exhaustion persists after weight loss; 2) Although abundant, Monocytes do not differ in transcriptional profile among these groups while dendritic cells and macrophages appear to change their activation status in response to obesity. Dataset generated from this work is a rich resource for identifying gene targets for focused investigations and hypothesis generation. Furthermore, an open-access online interactive portal is created by the authors to facilitate discovery and to broaden accessibility of this data, which is of value and interest to investigators in this research field. The design of this study is straightforward and the manuscript is well-written. However, this reviewer feels a few clarifications are needed:

1. 5' kits were used in this study yet no VDJ data is presented. did the authors do TCR analysis? if so, did T cell clonality increase as they previously reported mentioned in discussion?
2. it appears that ADTs data were under utilized and were only used for validation of clustering by gene expression, have the authors performed "weighted-nearest neighbor" analysis (PMID: 34062119) to improve the accuracy of clustering? also, did the authors confirm CD8 exhaustion by examining surface PD-1 expression using ADT?
3. the authors stated "a total of 33,322 cells that met strict quality control metrics (see Methods) were retained and integrated." however, it is unclear how the doublets were removed from their analysis, have the authors compared more than one doublet identification methods? initially, 5K cells were targeted per biological replicate. This less than 50% recovery rate appeared to be low and is this rate consistent among all biological replicates?
4. it unclear if all the replicates were process on the same day/run. did the authors observe or run batch effect correction?
5. this is a minor issue but a better cell hashing strategy could be pooling 4 mice from different diet group into one GEM.

Reviewer #3 (Remarks to the Author):

This manuscript by Cottam and colleagues presents intriguing data comparing the effects of weight loss and weight cycling on adipose tissue immune cell profiles. Overall, the data are interesting and compelling, and will significantly contribute to the field. Suggestions to improve the manuscript are outlined below.

Of note, I was specifically asked to provide insight on the weight-cycling/ weight-regain aspects of this manuscript. Thus, I will defer to other reviewers regarding the other aspects of the manuscript.

Major items:

While I find this data fascinating, the biggest issue I have with the manuscript/study is that I'm left wondering how the most recent diet impacts all of the outcomes (ie, eating a 60% vs 10% fat diet for the last 9 weeks of the study). Diet composition is known to affect adipose biology. While this cannot be addressed with the current study design, it would be interesting to examine the same outcomes when weight loss and/or weigh cycling are conducted with all animals consuming the same diet (ie calorically restrict the animals by adjusting the amount of food provided, rather than the diet composition). My assumption is that this data is not available. However, perhaps the authors have some additional data at each of the diet switch time points that might be a first step to addressing this issue. In addition, differences in lean mass, could also affect many of the outcomes in this study. Thus, statistical analyses could be conducted to test this potential effect. Regardless, additional discussion of these issues requires at least a paragraph in the discussion of this paper.

In addition, it is imperative to include some discussion of the fact that this entire study was performed in males, and this needs to be made clear throughout the manuscript (rather than just in the methods). There is a growing body of evidence to suggest that males and females differ in many aspects of weight-regain and/or adipose biology.

Insulin resistance: Given the difference in baseline plasma glucose in the GTT (Fig 1a), it would be worthwhile to include additional measures of whole-body metabolic health / insulin sensitivity.

Have the authors measured fasting insulin at the same timepoint as the GTT baseline?

Additionally, do you have measures of body composition and glucose/insulin at the end of each 9-week diet phase? It is likely that these differences are reflective of the last diet consumed – ie the obese and weight cycled had the HFD for the 9 weeks prior to the GTT; thus, it makes sense that their baseline plasma glucose is higher and that their AUC for glucose is higher. Similarly, the WL group has higher lean mass, and also lower plasma glucose AUC. Knowing that skeletal muscle beneficially impacts insulin sensitivity, perhaps the difference in lean mass accounts for some of this difference. For these reasons, any additional analyses (measures at different time points, correlations, covarying for lean mass to see if the differences between groups disappears, etc) to tease some of this apart would be helpful. It would also be helpful to include graphs of the fat pads/lean mass/liver as % of body weight as a supplemental figure.

I also recommend additional discussion that place this work in the context of what is known about changes in adipose physiology following weight-cycling and/or weight-regain. There is a grown body of literature in this area which seems to be left out of the discussion. Investigators such as Dulloo come to mind. The paper would have increased impact if the results could be placed in the context of "how" these changes associated with weight-cycling might predispose patients to future disease risk. This is done very briefly on the bottom of p.13, but could be expanded.

Minor Issues:

Figure 1i – lipid droplet diameter - is a bit hard to interpret with the staggered lines. Could you just make these a line graph (without shading under the line) and place them on top of each other? The other question related to this is how lipid droplet may be different than adipocyte diameter (which could be similarly measured on an H&E image). Are you getting any small lipid droplets that are outside the cells, thus making these two slightly different measures?
Minor items

Fig 1C: Avg is more standard abbreviation than Avrg

Figure 3 legend – is Fig 3C showing fold change in Obese vs Lean? (or is this lean vs obese). Indicating directionality in the legend will be helpful for the reader. Right now it just states difference in lean and obese; if this is a fold change, the reader needs to know the group for which it's increased (or decreased)

p.7 – "generated a module containing multiple established features of T cell exhaustion: Pcd1, Tox, Entpd1, Tigit, and Lag3". Please provide reference(s) for this. Same for the other makers in the paper. (ie description of Fig 4 in the text) – phenotype markers, t-cell markers, etc; monocyte subsets (p. 8) etc.

Lipid associated macrophages: are these different than metabolically activated macrophages (ie PMID 28954231), which seem to be macrophages activated by free fatty acids, that also take up lipids. Given the variations in macrophage nomenclature used across studies, it would be helpful to clarify.

p.10 – last paragraph before the discussion – the authors state that obesity shifted macrophages towards a more pro-inflammatory phenotype. It would be helpful to either indicate on the figure and figure legend which direction is more inflammatory (ie, towards M1 like, or a higher MPI number); conversely the text could be updated on p. 10 to clarify this issue. This would make the paper more accessible to those who are not as familiar with the field.

RESPONSE TO REVIEWER COMMENTS

We thank the Reviewers for the time and effort spent in carefully reviewing our manuscript and providing constructive comments. We are pleased with their overall enthusiasm and have worked hard to address all of the reviewer's concerns. We hope that the reviewers find this revised version of the manuscript to be acceptable for publication. Point-by-point responses to all comments and modifications to the manuscript are listed below. *Italicized text indicates text that is changed in the revised manuscript.*

Reviewer 1:

The study describes the dynamics of adipose tissue immune cell populations in a mouse model of weight cycling (WC). There is an epidemiological rationale, which relies on the fact that in human obesity up to 60% of patients undergoing body mass reduction, will eventually present body mass regain; there is also a molecular/cellular rationale, which relies on the fact that adipose tissue inflammation provides an important mechanistic basis for insulin resistance and also for a number of other obesity comorbidities.

Here, authors employed CITE-seq to explore the putative changes in immune cell populations in the adipose tissue in body mass gain, body mass reduction and WC. They show that cellular markers of antigen presentation, T-cell exhaustion, lipid handling and inflammation persist after body mass reduction and worsen with body mass regain.

In general, this is a fine and timely study that provides methodological advance in the way we can study adipose tissue inflammation; it also provides huge amount of data that can be further explored by the group and also by others, in order to provide an in-depth analysis of the adipose tissue inflammation in obesity. Finally, and most importantly, the study provides important advance in the understanding of adipose tissue inflammation in obesity, showing that WC can worsen the immune phenotype of the adipose tissue infiltrates, which directly impact on the worsening of the metabolic phenotype.

Major issues

1. Authors should provide an expanded description of the CITE-seq method.

RESPONSE: We have expanded on our description of the CITE-seq method in the Results (**page 6**): *“CITE-seq antibodies were used to confirm and improve cell annotation as follows: T cells (CD3, CD4, CD8 α), TCR γ/δ), NK cells (NK1.1), B cells (CD19), myeloid cells (CD11b), macrophages (FC γ R1, MAC2/GAL3), DCs (CD11c), and neutrophils (CD39) as well as costimulation and activation/inhibition markers (CD279/PD-1, TIGIT, CD44, CD80, CCR7/CD197) (Supplemental Table 1).”*

2. In page 5, authors say that a total of 33,322 cells met quality control. What was the number of cells per group? Was there any statistical difference in the number of cells obtained per group?

RESPONSE: There is a statistical difference in number of cells obtained per diet group. However, all data shown is normalized to the number of cells analyzed per mouse to correct for differences in total cell numbers. The number of cells per group are as follows: Lean – 5892, Obese – 8511, WL – 8846, WC – 10,073. Supplemental Fig 5 has been updated to show the exact number of cells per group, the proportion for each of the 4 mice, and the text referring to

this figure has also been updated (**page 6**): Cell types were well represented in all biological replicates without any major outliers driving our interpretation “and cell classification (Supplemental Fig. 5a), and all data shown are normalized to the number of cells analyzed per mouse to correct for differences in total cell numbers. Across the four sample groups (16 mice), a total of 33,322 cells that met strict quality control metrics (see Methods) were retained and integrated (Supplemental Fig. 5b).”

3. The WC model is interesting; however, it reflects the consequences of one cycle, only. In human obesity, patients frequently undergo several cycles of body mass reduction and regain. This should be acknowledged in the Discussion.

RESPONSE: This is a great point, we have added this to the “Considerations” section at the end of the Discussion (**page 17**): “This work represents only one cycle of WL and regain and utilizes a switch from primarily high fat to low fat feeding, which likely differs from human WC. Others have shown that multiple weight change cycles worsen glucose tolerance⁶⁹ (similar to our model)...The immune profile of WC mice was not evaluated in any of these models published by other groups, and it is likely that models with greater weight gain across multiple cycles would show even more immunological difference.”

4. There is no description of the method and purpose of performing partition-based graph abstraction (PAGA). This should be included.

RESPONSE: Partition-based graph abstraction (PAGA) utilizes the manifold produced by RNA velocity in combination with the connectivity of cell clusters in our dimensional reduction. In some cases, such as for CD8⁺ T cells, we felt that this method improved visualization by reducing noise considering the sparsity of the data. On the other hand, we found that for dendritic cells, PAGA predicted connections between biologically exclusive nodes (such as for cDC1 and cDC2 clusters). We believe that this is do to convergent RNA splicing events in phenotypically different cell groupings (for example, cDC1s and cDC2s both becoming activated). Ultimately, we decided that showing only the RNA velocity embedding (instead of PAGA visualization) did not change our conclusions for CD8⁺ T cells, but improved the clarity of the manuscript. However, we have included the original PAGA visualization within the associated vignettes should others be interested in the approach to generate it.

5. Monocyte recruitment to adipose tissue during the development of obesity is an important issue in this field. In their models, authors found no differences in the monocyte phenotypes; however, this seems odd. Authors could go deeper into the investigation of monocyte phenotypes looking into chemokine markers.

RESPONSE: Further analysis did reveal differences in lipid handling, activation/adhesion, and co-stimulatory genes. We’ve added these results (and discussed cell number in relation to the literature) in the results (**page 9**): “While monocyte recruitment to the adipose tissue is observed with obesity, population changes are time-dependent and often masked by large changes in the proportion of other cell types^{18,19}. Upon further assessment of cytokine, chemokine, and other functional markers, there were few differences in non-classical monocytes, but we did observe an increase in genes associated with lipid handling (*Trem2*, *Cd36*, *Cd9*)¹⁸, activation/adhesion (*Cd9*, *Cd81*, and *Cd63*)^{36,37}, and co-stimulation (*Cd86*, *Cd40*)³⁸ which were not reversed with weight loss in the classical monocyte subset (Fig. 5d). While *Cd86* gene expression increased following obesity, no change in *Cd80* mRNA or protein expression was observed due to diet within the classical monocyte subcluster.”

Additionally, the change in lipid handling supports the LAM RNA velocity data (**page 11**): the majority of LAMs are likely derived from tissue infiltrating monocytes, as previously suggested “that acquire features of lipid handling prior to differentiation (Fig. 5d).”

Finally, the changes in adhesion related genes led us to include this interesting discussion point (**page 14**): “Interestingly, the tetraspanins CD9, CD63, and CD81 which were increased in classical monocytes with weight cycling have been suggested to play a role in multinucleated giant cell formation⁵⁹. Giant multinucleated cells have been found in obese adipose tissue and contribute to the clearance of dead adipocytes^{60,61}. However, in our studies, these large cells were likely filtered out during cell isolation, and thus it is not known if they change with WC.”

Minor issues

1. There is a typo on subsection title Body composition and glucose tolerance (page 15).

RESPONSE: Thank you. This typo has been corrected.

2. In page 11, the sentence – We postulate that this obesity-associate immunophenotypic imprinting... - is speculative as the study has neither evaluated cardiovascular outcomes nor the relation of the adipose tissue immune cells with putative cardiovascular abnormalities.

RESPONSE: Thank you for this critique. While we are personally very interested to know if similar impacts are observed with cardiometabolic disease in mouse models of weight cycling, we have corrected this statement to read “*metabolic health*” to fit within the scope of this manuscript and our results (**page 12**).

Reviewer 2:

In this study, Cottam et al successfully developed a mouse model representing weight loss-accelerated metabolic disease then performed CITEseq analysis on 33,322 immune cells in the adipose tissues from a total of 16 mice assigned into 4 diet groups. Major findings from these analyses include 1) Obesity-associated T cell exhaustion persists after weight loss; 2) Although abundant, Monocytes do not differ in transcriptional profile among these groups while dendritic cells and macrophages appear to change their activation status in response to obesity. Dataset generated from this work is a rich resource for identifying gene targets for focused investigations and hypothesis generation. Furthermore, an open-access online interactive portal is created by the authors to facilitate discovery and to broaden accessibility of this data, which is of value and interest to investigators in this research field. The design of this study is straightforward and the manuscript is well-written.

However, this reviewer feels a few clarifications are needed:

1. 5' kits were used in this study yet no VDJ data is presented. did the authors do TCR analysis? if so, did T cell clonality increase as they previously reported mentioned in discussion?

RESPONSE: TCR analysis was conducted and T cell clonality does indeed increase with obesity as we have previously shown. Continued analysis of this data is in progress and is a

critical component of a subsequent paper (currently in draft) focused on T cell function in weight loss and weight cycling, which is beyond the scope of the current manuscript.

2. It appears that ADTs data were under-utilized and were only used for validation of clustering by gene expression, have the authors performed “weighted-nearest neighbor” analysis (PMID: 34062119) to improve the accuracy of clustering? Also, did the authors confirm CD8 exhaustion by examining surface PD-1 expression using ADT?

RESPONSE: We originally did test weighted-nearest neighbor clustering prior to cell annotation, but did not observe improvement in cluster accuracy compared to careful annotation of gene expression clusters. In some cases, absence of a discriminating surface marker for a specific cell type caused cells to be clustered more ambiguously with the weighted-nearest neighbor approach. For example, absence of the gamma/delta surface marker (which was not included in our panel of CITE-seq antibodies) resulted in these cells sometimes being clustered with alpha/beta T cells due to presence of the CD3e surface marker.

The surface PD-1 (CD279) and TIGIT markers do confirm CD8⁺ T cell exhaustion in our data. We have updated the text to indicate this point and have included an additional panel to Figure 4 highlighting these CITE-seq proteins (**page 9**): “*This was further confirmed by a T cell exhaustion module based on protein expression of PD-1 (CD279) and TIGIT from our CITE-sequencing antibodies (Fig. 4g).*”

3. The authors stated "a total of 33,322 cells that met strict quality control metrics (see Methods) were retained and integrated." however, it is unclear how the doublets were removed from their analysis. Have the authors compared more than one doublet identification methods? Initially, 5K cells were targeted per biological replicate. This less than 50% recovery rate appeared to be low and is this rate consistent among all biological replicates?

RESPONSE: Our doublet detection approach relied on hashtag antibodies, but it is possible that cells from the same animal (same hashtags) were captured together. To address this concern, we utilized the DoubletFinder R package, which simulates doublet cells and identifies which cells cluster with them. Importantly, DoubletFinder can only reliably estimate heterotypic doublets (doublets containing cells of different types). This tool identified 311 potential doublets (~0.93% of cells). Upon further investigation, the majority of doublets were annotated as either proliferating dendritic cells or stromal cells. For dendritic cells, differential expression analysis identified only 4 statistically significant genes with a marginal log₂ fold change (>0.5) of which none were cell type specific: Cdk1, Top2a, Tk1, and Rrm2. Therefore, we do not believe these to be true doublets. For stromal cells, 157 differentially expressed genes were identified comparing singlets to doublets. Some of the identified genes suggest there may be heterotypic doublets of stromal cells with macrophages, mast cells, and monocytes.

We have updated the considerations section with a reference containing much more expansive data on the adipose tissue stromal compartment and noted this limitation in our data set (**page 18**): “*Finally, our analysis was largely focused on the immune cell compartment of adipose tissue. However, we identified 1,836 cells that were annotated as stromal cells despite magnetically sorting for CD45. Doublet detection using DoubletFinder indicated approximately 10.2% of these cells may be heterotypic doublets (compared to only 0.28% for all other clusters combined). Differential gene expression identified genes associated with macrophages, mast cells, and monocytes in stromal cells labeled as doublets compared to those labeled as singlets. Therefore, studies focused on adipocyte progenitors⁷⁷ or mature adipocytes^{78,79,80} (captured via*

single nuclei sequencing) may provide more insight into the relationship between stromal and immune cells and could be expanded to include cells from WC mice.”

The numbers of predicted doublets are shown below:

DoubletFinder_V3			
Cell Type	Singlet	Doublet	% Predicted Doublets
Macrophages	11432	39	0.34
Monocytes	3245	2	0.06
Dendritic Cells	4527	97	2.14
T Cells	4658	1	0.02
NK Cells	2383	0	0.00
B Cells	2560	0	0.00
Plasma Cells	664	1	0.15
Mast Cells	1039	1	0.10
Neutrophils	201	0	0.00
ILCs	636	0	0.00
Stromal Cells	1666	170	10.20

Prior to quality control, we obtained 63,793 cells (originally targeting a total of 80,000 cells). Many of these cells were removed by strict quality control parameters. Additionally, we found that surface tags often had some noise, which resulted in many cells that had ambiguous or absent hashtags. Our study was originally designed to work around this problem, since each lane corresponded to only one biological group (with four replicates). However, we felt that it was important to only retain cells that we could confidently link back to the original biological replicate.

4. It unclear if all the replicates were process on the same day/run. Did the authors observe or run batch effect correction?

RESPONSE: All replicates were processed on the same day and run, so minimal batch effects were expected. We have clarified the methodology to indicate this important point (**page 21**): All samples were submitted and processed for sequencing “*on the same day to minimize batch effects.*”

The reciprocal PCA method incorporated into Seurat does perform batch correction due to utilization of variable integration features. We also tested numerous other integration methods, such as Harmony and scTransform. However, we found that these methods resulted in a more liberal batch correction. Specifically, scTransform was found to overcorrect in our data, resulting in cluster markers that were unlikely to be mutually exclusive (e.g. genes coding for ribosomal proteins that are highly expressed in all cell types).

5. This is a minor issue but a better cell hashing strategy could be pooling 4 mice from different diet group into one GEM.

RESPONSE: We agree that this approach better controls for biological variability between replicates and debated running our samples this way. The suggested approach requires very accurate sample demultiplexing since distinct biological groups are pooled. Therefore, were

originally concerned that absence of clear hashtag signal would render the data unusable, whereas our approach would have still retained distinct groups, even if we did not obtain discernable biological replicates.

Reviewer 3:

This manuscript by Cottam and colleagues presents intriguing data comparing the effects of weight loss and weight cycling on adipose tissue immune cell profiles. Overall, the data are interesting and compelling, and will significantly contribute to the field. Suggestions to improve the manuscript are outlined below. Of note, I was specifically asked to provide insight on the weight-cycling/ weight-regain aspects of this manuscript. Thus, I will defer to other reviewers regarding the other aspects of the manuscript.

Major items:

1. While I find this data fascinating, the biggest issue I have with the manuscript/study is that I'm left wondering how the most recent diet impacts all of the outcomes (ie, eating a 60% vs 10% fat diet for the last 9 weeks of the study). Diet composition is known to affect adipose biology. While this cannot be addressed with the current study design, it would be interesting to examine the same outcomes when weight loss and/or weigh cycling are conducted with all animals consuming the same diet (ie calorically restrict the animals by adjusting the amount of food provided, rather than the diet composition). My assumption is that this data is not available. However, perhaps the authors have some additional data at each of the diet switch time points that might be a first step to addressing this issue.

RESPONSE: The reviewer raises an important issue with regards to the weight cycling model. The weight cycling design used in our studies has mice switch from 60% fat (duration=9 weeks) to 10% fat (weight loss, duration=9 weeks) then back to 60% fat during weight regain (duration=9 weeks). The reviewer is correct that we do not have scRNA seq data on mice that have lost weight due to calorie restriction while maintaining HFD. However, we have another study underway that addresses some of the diet composition questions. These data are being compiled for another manuscript focused on uncovering the mechanisms by which weight cycling worsens glucose homeostasis. We have included some of these unpublished data below, to address specific inquiries from the reviewer. We hope these data address the reviewer's comment; however, we feel that because we do not have CITE-seq data for these mice, they would detract from the overall message of the current manuscript.

To address whether diet composition influences metabolic health following weight loss and subsequent weight cycling, a group of mice were pair fed (PF) to the weight cycled group during the weight loss phase (*ad lib* HFD_{0-9 wk} → HFD kcal restricted_{10-18 wk} → *ad lib* HFD_{19-27 wk}) to match body weight loss. As noted in the figure diagram the pair fed group had calories restricted only during the weight loss phase (9-18 weeks) of the study. The remainder of the experiment these mice were fed *ad lib*. Body weight curves, energy intake, and body composition between weight cycled and weight cycled-PF groups were not different (**Panels B-E**). In addition, there were no differences in fasting glucose, or insulin concentrations following the weight loss period or the weight regain period in weight cycled with LFD versus weight cycled-PF animals, respectively (**Panels F&G**). These data indicate that both weight cycling groups were well-matched, but also reveal that consuming diets with different macronutrient (high fat versus low fat) composition during the weight loss phase did not differentially alter fasting indices of glucoregulation after 9 weeks of weight loss or weight regain.

To address this question in the manuscript, we have generated a new section in the Discussion entitled “Considerations” to discuss a number of the issues raised by the reviewer (page 16-17), including “As diet composition can affect adipose biology⁷⁵, models using caloric restriction or altered diet composition, as well as exercise, pharmacological, bariatric surgery, or environmental temperature would greatly improve our understanding of WC-associated disease.”

2. In addition, differences in lean mass, could also affect many of the outcomes in this study. Thus, statistical analyses could be conducted to test this potential effect. Regardless, additional discussion of these issues requires at least a paragraph in the discussion of this paper.

RESPONSE: The reviewer raises the point that differences in lean mass between the 'lean' group and 'weight loss' group may contribute to some of the metabolic responses such as glucose tolerance. ANCOVA analyses were conducted to statistically account for lean mass in GTT AUC. When lean mass is covaried against GTT AUC for all groups (Lean, obese, WC, and WL), the statistical effects on GTT AUC is maintained (i.e., lean mass did not explain glucose clearance between groups). This can be visualized by the estimated marginal means displayed below. In addition, since lean mass was statistically higher in WL versus lean animals and WL mice had greater glucose clearance than lean mice, lean mass was covaried against GTT AUC with only the lean group and WL group included in the model. Similar to the full model, GTT AUC was still significantly lower in WL versus lean mice after accounting lean mass. Thus, it does not appear that the difference in lean mass is the driving force for differences in glucose excursions among groups. It is worth noting that although lean mass is statistically significantly different between lean and weight loss mice, the mean difference is ~1 g. It is unlikely that this absolute difference in mass accounts for greater glucose clearance. We have clarified the differences in the results section the discussion as follows (**page 4-5**): *"Fat free tissue comprises the bulk of insulin-stimulated glucose disposal and is positively associated with postprandial glucose clearance^{22,23}. Given that lean mass was greater in WL versus lean mice and WL had greater glucose clearance than lean mice, lean tissue mass was covaried against GTT AUC. The decrease in glucose clearance in WL compared to lean animals manifested after statistically accounting for differences in lean mass."*

3. In addition, it is imperative to include some discussion of the fact that this entire study was performed in males, and this needs to be made clear throughout the manuscript (rather than just in the methods). There is a growing body of evidence to suggest that males and females differ in many aspects of weight-regain and/or adipose biology.

RESPONSE: We have made this important sex distinction throughout the manuscript (added to Abstract, Introduction, Results, and Discussion). We additionally include female data (**page 5**, Supplemental Fig. 1), and we discuss sex dimorphism in weight gain and adipose biology as it pertains to this work in the "Considerations" section of the Discussion. In female mice, glucose AUC was not statistically different between WC and obese animals, yet glucose clearance

during the excursion was modestly delayed in WC versus obese females. This suggests that female mice are likely on a path towards worsened metabolic control with weight cycling. We hypothesize that if female mice initiated the study at an older age when body weight gain is steeper on HFD, the subsequent weight cycling phenotype (i.e., augmented glucose intolerance) would manifest; however, we have not yet tested this hypothesis.

Results (**page 5-6**): *“We also determined whether weight cycling worsened metabolic control in female mice. At the end of the 27-week study, body weight, lean mass, and fat mass were greater in both obese and WC females than lean controls, whereas no differences were detected between obese vs WC groups. In contrast to male mice, weight cycling did not significantly exacerbate glucose intolerance in female animals (i.e., glucose AUC); however, there was a modest delay in glucose clearance during the glucose excursion between WC and obese females (Supplemental Fig. 3). Together, these data demonstrate that our mouse model provides a robust representation of WC-accelerated metabolic disease in male mice, which were used for all subsequent experiments.”*

Considerations (**page 16**): *“Several aspects of this study require additional consideration. First, only male mice were included in the immunological studies; thus, we are unable to determine whether the observed changes in the adipose immune compartment linked with WL and WC manifest similarly in females. It is well established that female rodents are less susceptible to diet-induced obesity than males and display a different inflammatory phenotype in adipose tissue than males. Moreover, females have greater adipose and systemic insulin sensitivity than males for a given body mass^{64,65,66,67}, and the response to caloric restriction and subsequent hyperphagia following ad libitum food access is lower in female mice⁶⁸. Thus, as expected, female mice in this study gained less weight than males on HFD. Weight cycling females did not markedly worsen glucose tolerance, however their initial bout of weight gain following 9 weeks of HFD was minimal. We hypothesize that if female mice initiated the study at an older age when body weight gain is steeper on HFD, the subsequent weight cycling phenotype (i.e., augmented glucose intolerance) would manifest. It is also likely that greater weight fluctuations are required to worsen metabolic function in females. Nonetheless, it is probable that at least in this model, WC females would induce distinct immune remodeling compared with males. Future studies could use different models (such as starting diet at an older age when body weight gain is steeper or ovariectomizing the mice) to assess the differences in male and female responses to WC and the degree of weight variability required to observe metabolic differences.”*

4. Insulin resistance: Given the difference in baseline plasma glucose in the GTT (Fig. 1a), it would be worthwhile to include additional measures of whole-body metabolic health / insulin sensitivity. Have the authors measured fasting insulin at the same timepoint as the GTT baseline?

RESPONSE: We assessed fasting insulin in a subset of obese and weight cycled mice. We did not detect differences in fasting insulin between obese and weight cycled mice. These new data have been added to the results as follows (**page 4**): *“Fasting insulin concentrations were not different between obese and WC animals (Obese, 5.2 ± 0.8 ng/ml and WC, 5.1 ± 0.9 ng/ml, $p=0.9$).”*

5. Additionally, do you have measures of body composition and glucose/insulin at the end of each 9-week diet phase? It is likely that these differences are reflective of the last diet consumed – ie the obese and weight cycled had the HFD for the 9 weeks prior to the GTT; thus, it makes sense that their baseline plasma glucose is higher and that their AUC for glucose is higher.

RESPONSE: Although insulin was not measured at interim time points, we determined both body composition and glucose tolerance for male mice in past experiments at 3, 9, and 18 weeks. We observe impairment in glucose tolerance starting as early as 3 week and increasing at 9 weeks of high fat diet. Mice switched from high fat diet to low fat diet (9 weeks period) show complete recovery of glucose tolerance prior to weight cycling. We have included the following data in an additional supplemental figure to address this important consideration (**page 5**): *“Effects of HFD feeding and weight loss (between 9 and 18 weeks of the intervention) on body composition and glucose tolerance are reported in the data supplement (Supplemental Fig. 2)”*.

6. Similarly, the WL group has higher lean mass, and also lower plasma glucose AUC. Knowing that skeletal muscle beneficially impacts insulin sensitivity, perhaps the difference in lean mass accounts for some of this difference. For these reasons, any additional analyses (measures at different time points, correlations, covarying for lean mass to see if the differences between groups disappears, etc) to tease some of this apart would be helpful.

RESPONSE: The reviewer is correct that fat free tissue accounts for the bulk of insulin mediated glucose disposal. For these reasons, glucose tolerance tests were dosed based upon lean mass. However, it is possible that higher lean mass in the WL group could contribute to greater glucose clearance than lean controls during a glucose challenge. As noted above in **point #2** above, statistically accounting for lean mass (ANCOVA) did not explain differences in glucose AUC between WL and lean mice. Of note, lean mass is not different between obese and weight cycled mice and is therefore unlikely to explain worsened glucose tolerance. We cannot exclude the possibility that local muscle insulin action is worsened in weight cycled versus obese mice. This latter question is currently being addressed in another study using hyperinsulinemic-euglycemic clamp technique coupled with isotopic tracing methodologies.

7. It would also be helpful to include graphs of the fat pads/lean mass/liver as % of body weight as a supplemental figure.

RESPONSE: We have added these data as Fig. 1h.

8. I also recommend additional discussion that place this work in the context of what is known about changes in adipose physiology following weight-cycling and/or weight-regain. There is a grown body of literature in this area which seems to be left out of the discussion. Investigators such as Dulloo come to mind. The paper would have increased impact if the results could be placed in the context of “how” these changes associated with weight-cycling might predispose patients to future disease risk. This is done very briefly on the bottom of p.13, but could be expanded.

RESPONSE: We have included a brief mention of the other findings (specifically adipose and metabolic related) of WC studies (**page 17**): *“Others have shown that multiple weight change cycles worsen glucose tolerance (similar to our model). Moreover, other WC models have demonstrated that WC reduces adiponectin and CRTP^{70,71}, downregulates clock genes⁷², and increases fat regain due to the loss of lean mass with WL which increases appetite, reduces energy expenditure, and reduces adaptive thermogenesis^{69,73,74}. The immune profile of WC mice was not evaluated in any of these models, and it is likely that models with greater weight gain across multiple cycles would show even more immunological difference.”*

Regarding how these changes are linked to WC, we did not want to speculate too far, as the exact role of many of these cell types and functions in regulating adipose homeostasis during weight gain are not well understood in obesity, much less WC. However, we agree that this is the primary rationale for the research, and thus we have a few more sentences in the Discussion (**page 15**): *“Unfortunately, the role of many of these cell types and functions in regulating adipose homeostasis during weight gain are not well understood, and even less is known in WL and WC. However, our results suggest critical areas of interest for future studies.”*

Minor Issues:

9. Figure 1i – lipid droplet diameter - is a bit hard to interpret with the staggered lines. Could you just make these a line graph (without shading under the line) and place them on top of each other? The other question related to this is how lipid droplet may be different than adipocyte diameter (which could be similarly measured on an H&E image). Are you getting any small lipid droplets that are outside the cells, thus making these two slightly different measures?

RESPONSE: We have updated Figure 1I to improve interpretation of the adipocyte size quantification.

Additionally, we decided to use lipid droplet diameter (staining for Perilipin-1) because the lipid droplet in mature adipocytes takes up nearly the entire volume of the cell. Importantly, Perilipin-1 immunolabeling has a distinct advantage over H&E because dying adipocytes (found in crown-like structures) do not express PLIN-1 and therefore only viable cells are included in measurements of adipocyte size. We did not observe any small lipid droplets outside of cells by PLIN-1 immunolabeling. Importantly, we have clarified in our Methods section (**page 20**): *“Only cells with a measured diameter greater than 10 microns were included in our analysis.”*

Minor items:

10. Fig 1C: Avg is more standard abbreviation than Avg

RESPONSE: Thank you, this abbreviation has been corrected in Fig. 1c and Supplemental Fig. 1c.

11. Figure 3 legend – is Fig 3C showing fold change in Obese vs Lean? (or is this lean vs obese). Indicating directionality in the legend will be helpful for the reader. Right now it just states difference in lean and obese; if this is a fold change, the reader needs to know the group for which it's increased (or decreased)

RESPONSE: These results are obese vs. lean, as in Fig. 3b. We have clarified this in the figure legend: *“From obese to lean mice”*.

12. p.7 – “generated a module containing multiple established features of T cell exhaustion: Pdcd1, Tox, Entpd1, Tigit, and Lag3”. Please provide reference(s) for this. Same for the other makers in the paper. (ie description of Fig 4 in the text) – phenotype markers, t-cell markers, etc; monocyte subsets (p. 8) etc.

RESPONSE: We have added references for use of specific subset markers for T cells, monocytes, dendritic cells, and macrophages.

13. Lipid associated macrophages: are these different than metabolically activated macrophages (ie PMID 28954231), which seem to be macrophages activated by free fatty acids, that also take up lipids. Given the variations in macrophage nomenclature used across studies, it would be helpful to clarify.

RESPONSE: We agree that this terminology is similar, however, MMe is a polarization designation that describes general changes seen in bulk adipose macrophages from obese mice/ humans, while LAM are one specific population in obese adipose tissue. While some of these markers are published to be increased in both populations (*Plin2*, *Cd36*), not all markers have been compared (ex. *Trem2*). For clarity, we are using LAM as they fit one specific cluster of adipose macrophages. It would be useful for future research to better evaluate the overlap between MMe and LAM markers and functional changes throughout the adipose tissue macrophage compartment.

14. p.10 – last paragraph before the discussion – the authors state that obesity shifted macrophages towards a more pro-inflammatory phenotype. It would be helpful to either indicate on the figure and figure legend which direction is more inflammatory (ie, towards M1 like, or a higher MPI number); conversely the text could be updated on p. 10 to clarify this issue. This would make the paper more accessible to those who are not as familiar with the field.

RESPONSE: Thank you for noticing this, we have added clarity to the figure legend and results statement (**page 12**): *“We observed that obesity shifted both subpopulations of macrophages towards a more pro-inflammatory phenotype that was not recovered following WL, indicated by a higher MPI which signifies gene expression patterns associated with M1-like phenotypes.”*

Again, we would like to thank all three reviewers for their thoughtful critiques. We feel the manuscript is much better after these modifications, and hope the reviewers and editors will now find it acceptable for publication.

Reviewer comments, further review

Reviewer #1 (Remarks to the Author):

Revised version has improved consistently. I have no further comments.

Reviewer #2 (Remarks to the Author):

My concerns have been addressed in the revision

Reviewer #3 (Remarks to the Author):

I appreciate the effort that the authors made in revising the manuscript. All of my original concerns have been address adequately.